# Generation of patterned kidney organoids that recapitulate the adult kidney collecting duct system from expandable ureteric bud progenitors

Zipeng Zeng[1,2,4], Biao Huang[1,2,4], Riana K. Parvez[2], Yidan Li[1,2], Jyunhao Chen[1,2], Ariel C. Vonk[1,2], Matthew E. Thornton [3], Tadrushi Patel[1,2], Elisabeth A. Rutledge [2], Albert D. Kim[2], Jingying Yu[1,2], Brendan H. Grubbs [3], Jill A. McMahon[2], Nuria M. Pastor-Soler[1], Kenneth R. Hallows[1], Andrew P. McMahon[2] & Zhongwei Li [1,2 ✉]

Current kidney organoids model development and diseases of the nephron but not the contiguous epithelial network of the kidney's collecting duct (CD) system. Here, we report the generation of an expandable, 3D branching ureteric bud (UB) organoid culture model that can be derived from primary UB progenitors from mouse and human fetal kidneys, or generated de novo from human pluripotent stem cells. In chemically-defined culture conditions, UB organoids generate CD organoids, with differentiated principal and intercalated cells adopting spatial assemblies reflective of the adult kidney's collecting system. Aggregating 3D-cultured nephron progenitor cells with UB organoids in vitro results in a reiterative process of branching morphogenesis and nephron induction, similar to kidney development. Applying an efficient gene editing strategy to remove RET activity, we demonstrate genetically modified UB organoids can model congenital anomalies of kidney and urinary tract. Taken together, these platforms will facilitate an enhanced understanding of development, regeneration and diseases of the mammalian collecting duct system.

[1] USC/UKRO Kidney Research Center, Division of Nephrology and Hypertension, Department of Medicine, Keck School of Medicine, University of Southern California, Los Angeles, CA, USA. [2] Deptartment of Stem Cell Biology and Regenerative Medicine, Keck School of Medicine, University of Southern California, Los Angeles, CA, USA. [3] Division of Maternal Fetal Medicine, Keck School of Medicine, University of Southern California, Los Angeles, CA, USA. [4] These authors contributed equally: Zipeng Zeng, Biao Huang. ✉email: zhongwei.li@med.usc.edu

The mammalian kidney contains thousands of nephrons, connected to a highly branched collecting duct (CD) system. Nephrons filter and process the blood to form the primitive urine, which is collected and further refined in the CD system to adjust water, electrolytes, and pH and to maintain the homeostasis of tissue fluid[1,2]. The complex and elaborate kidney is largely formed from the reciprocal interactions of two embryonic cell populations: the epithelial ureteric bud (UB), and the surrounding metanephric mesenchyme (MM). Signals from the MM induce the repeated branching of UB, which gives rise to the entire CD network. Meanwhile, signals from the UB induce the MM to form nephrons[3,4]. Given this central role of the UB in kidney organogenesis, defects in UB/CD development often lead to malformation of the kidney, low endowment of nephrons at birth, and congenital anomalies of kidney and urinary tract (CAKUT)[2,4,5]. Thus, a better understanding of kidney branching morphogenesis is needed for in vitro efforts toward rebuilding the kidney. It is also required for developing novel preventive, diagnostic, and therapeutic approaches for various kidney diseases.

Three-dimensional (3D) multicellular mini-organ structures, or organoids, have broad applications for modeling organ development and disease, and for regenerating organs through cell or tissue replacement therapies[6,7]. Recently, we and others have been able to generate kidney organoids from human pluripotent stem cells (hPSCs)[8–11] or from expandable nephron progenitor cells (NPCs)[12–14]. These organoids have greatly aided studies of the role of nephrons in kidney development and disease[15]. However, despite previous efforts toward the expansion or de novo generation of the immature UB relying on primary mouse/rat tissue[16–20], mouse embryonic stem cells[21–23], or hPSCs[23–29], we still lack a robust kidney organoid model that can generate and expand the UB progenitor cells (UPCs), and recapitulate the maturation and spatial patterning of the adult CD (see Supplementary Tables 1 and 2 for a side-by-side comparison of this study and published literature).

Here, we report the development of a 3D organoid model that mimics the full spectrum of kidney branching morphogenesis in vitro—from the expandable immature UB progenitor stage, to the mature CD stage. UB organoids derived from either primary UB progenitor cells or hPSCs are amenable to efficient gene editing, and have broad applications for studying kidney development, regeneration, and disease.

## Results

**Expanding mouse UB progenitor cells into 3D branching UB organoids.** We previously developed a 3D culture system for the long-term expansion of mouse and human NPCs, which can generate nephron organoids that recapitulate kidney development and disease[14,30]. UB branching morphogenesis is driven by another kidney progenitor population, the UPCs. UPCs are specified around embryonic Day 10.5 (E10.5), when the UB starts to invade the MM. UPCs disappear around postnatal Day 2 (P2), when nephrogenesis ceases. Self-renewing UPCs reside in the tip region of the branching UB. During their ~10-day lifespan, some UPCs migrate out of UB tip niche to the UB trunk, and differentiate into the renal CD network. Other UPCs proliferate and replenish the self-renewing progenitor cell population of the UB tip. Ret[31,32] and Wnt11[33] have been identified as specific markers for UPCs and regulate UPC programs directly (Ret) or through feedback mechanisms (Wnt11). A transgenic reporter mouse strain Wnt11-myrTagRFP-IRES-CE ("Wnt11-RFP" for short) facilitates the real-time tracking of Wnt11-expresing cells based on RFP expression, and the lineage tracing of their progeny via a Cre-mediated recombination system[34,35].

We employed this Wnt11-RFP reporter system as a readout to screen for a culture condition that maintained the progenitor identity of UPCs in vitro. T-shaped UBs were manually isolated from E11.5 kidneys of Wnt11-RFP mice, and immediately embedded into Matrigel to set up a 3D culture platform that supported epithelial branching. In this 3D culture format, built on prior efforts toward the ex vivo culture of UB[16–20], starting from the medium components used by Yuri et al.[20], hundreds of different combinations of growth factors and small molecules were tested, following strategies similar to those we had used to establish optimal NPC culture[14] (Supplementary Fig. 1a–l, see also the "Supplementary Methods" for details). This screening allowed us to identify a cocktail, which we named "UB culture medium" (UBCM, Supplementary Table 6), that maintained self-renewing UPCs as a 3D branching UB organoid (Fig. 1a). Under this culture condition, the T-shaped UB formed a rapidly expanding branching epithelial morphology. More importantly, in contrast to prior UB culture system that generated a mixture of both UB tip and trunk cell types[20], uniform Wnt11-RFP expression was maintained throughout the 3D structure in the UBCM-derived UB organoid, suggesting the capture of a relatively pure UPC population (Fig. 1b). Resected Wnt11-RFP$^+$ UB organoid tips, re-embedded in Matrigel, branched and grew into additional Wnt11-RFP$^+$ UB organoids. Repetitive passaging and embedding for up to 3 weeks, resulted in more than a hundred thousand-fold expansion in the number of cells (Fig. 1c). Wnt11-RFP levels remained uniform for the first 10 days but progressively dropped thereafter, similar to the normal time course of UPCs in vivo. Consistent with the uniform expression of Wnt11-RFP throughout UB organoids at 10 days, whole-mount immunostaining confirmed the homogenous expression of the critical UPC regulators Ret, Etv5, and Sox9, as well as broad UB lineage markers Gata3, Pax2, Krt8, and Cdh1 (Fig. 1d, e and Supplementary Fig. 2a, b).

To better define the identity of the UB organoids, we used RNA-seq to profile the transcriptome of the organoids after 5, 10, and 20 days in culture. These data were compared with prior RNA-seq data for primary UB tip and UB trunk populations[35], as well as for NPCs and interstitial progenitor cells (IPCs)[36]. Unsupervised clustering (Supplementary Fig. 2c) and principal component analysis (PCA) (Fig. 1f) placed the cultured UB organoids closer to the primary UB tip samples than to differentiated stalk derivatives of the UB trunk. Taken together, these findings indicate that the UB organoid culture system enables a substantial expansion of cells retaining molecular characteristics of UPCs in vitro.

Next, we tested whether the UB organoid culture system could be applied to mouse strains other than Wnt11-RFP. For this, we successfully derived UB organoids from E11.5 UB from various other mouse strains, such as Sox9-GFP[37] and Rosa26-Cas9/GFP[38] (Supplementary Fig. 2d, e). All of these UB organoids retained the typical branching morphology and showed very similar growth rates, compared to Wnt11-RFP UB organoids (Fig. 1g), indicating the robustness of the 3D/UBCM culture system. Importantly, UB organoids self-organized into branching organoids after a freeze–thaw cycle, enabling cryostorage and reseeding of UB cultures (Supplementary Fig. 2f).

To determine whether UBCM culture conditions enabled clonal growth from a single UPC, dissociated E11.5 UBs were embedded at clonal density in Matrigel and cultured in UBCM medium. Around 30% of the single cells self-organized into E11.5 UB-like budding structures within 5 days, though a smaller percentage (3–5%) maintained Wnt11-RFP (Fig. 1h, and Supplementary Fig. 2g, h), an efficiency similar to clonal organoid formation for Lgr5$^+$ intestinal stem cells[39]. Importantly, the

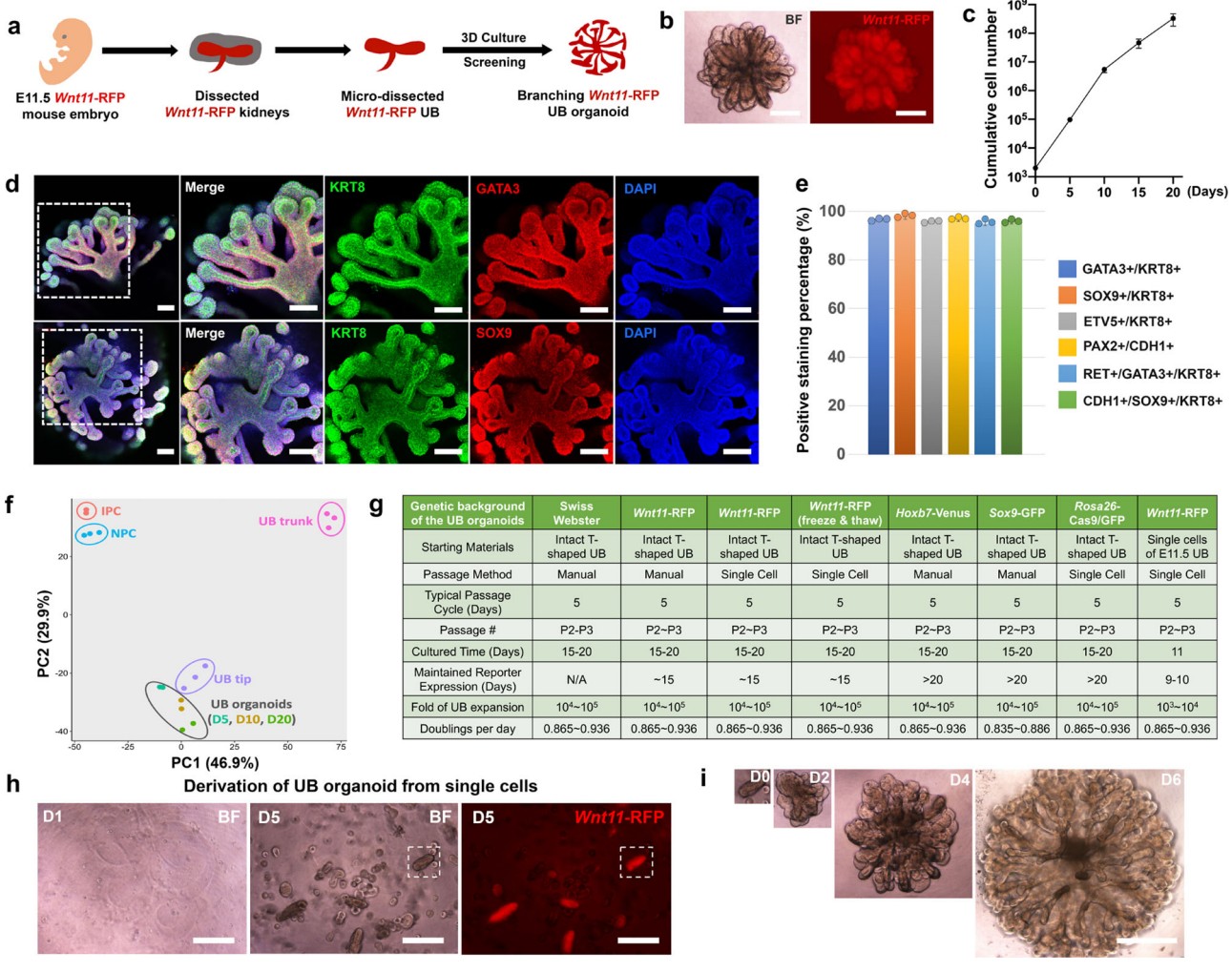

**Fig. 1 Expanding mouse UB progenitor cells into 3D branching UB organoids. a** Schematic of mouse UB isolation and screening for UB organoid culture condition. **b** Representative bright field (BF, left panel) and *Wnt11*-RFP (right panel) images of UB organoid. Scale bars, 200 μm. **c** Cumulative growth curve of UB organoid culture starting from 2000 cells. Each time point represents three biological replicates. **d** Whole-mount immunostaining of UB organoid for various UB markers at Day 10 of culture. The four panels on the right represent the boxed region in the left panel. Scale bars, 100 μm. **e** Quantification of percentages of UB cells stained positive for different UB markers in Fig. 1d and Supplementary Fig. 2a, b. Each column represents counts from three different fields of view (*n* = 3). **f** Principal component analysis (PCA) of RNA-seq data. Different colors and oval circles represent different primary kidney cell populations (NPC, IPC, UB tip, and UB trunk) or UB organoids cultured for 5 (D5), 10 (D10), and 20 days (D20). **g** Summary of UB organoid derivation from mouse strains with different genetic backgrounds or with different derivation methods. **h** Bright field (BF) images showing single UB cells, derived from *Wnt11*-RFP mice, cultured in the UBCM on Days 1 (left panel) and 5 (middle panel), as well as *Wnt11*-RFP image on Day 5 (right panel). Scale bars, 200 μm. **i** UB organoid derivation from a single UB cell-derived budding structure isolated from the boxed region in (**h**) at Days 0 (D0, the day of isolation and re-embedding into Matrigel), 2 (D2), 4 (D4), and 6 (D6). All images in (**i**) have been scaled to share the same scale bar with the D6 image. Scale bar, 400 μm. All data are presented as mean ± s.d. Source data are provided as a Source Data file.

clonally derived *Wnt11*-RFP⁺ budding structures were identical to intact E11.5 UB-derived organoids in both branching morphology and growth rate (Fig. 1g–i). Furthermore, withdrawal of the major medium components from UBCM resulted in either growth arrest (CHIR99021 and GDNF) or rapid loss of *Wnt11*-RFP (all components except for CHIR99021), suggesting that each component was essential for optimal UB organoid culture (Supplementary Fig. 1l). These data, taken together, suggest that UBCM represents a synthetic niche for the in vitro expansion of UPCs.

**Screening for conditions to mature UB organoids into CD organoids.** The functions of the mature renal CD system are carried out by two major cell populations that are intermingled throughout the entire CD network. The more abundant principal cells (PCs) concentrate the urine and regulate Na⁺/K⁺

homeostasis via water and Na⁺/K⁺ transporters. The less abundant α- and β-intercalated cells (ICs) regulate normal acid-base homeostasis via secretion of H⁺ or HCO3⁻ into the urine. The absence of an in vitro system recapitulating PC and IC development in an appropriate 3D context, constrains physiological exploration, disease modeling, and drug screening on the renal CD system. With this limitation in mind, we developed a screen to establish conditions supporting the differentiation of CD organoids, assaying expression of *Aqp2*[40] and *Foxi1*[41], definitive markers for PC and IC lineages, respectively, by quantitative reverse transcription PCR (qRT-PCR), following 7 days of culture under variable but defined culture conditions (Fig. 2a).

In a 1st round of screening, we determined the base condition in which minimal growth factors/small molecules sustained the survival of the organoids and permitted their differentiation. The base medium used for UBCM—hBI[14]—was tested, together with

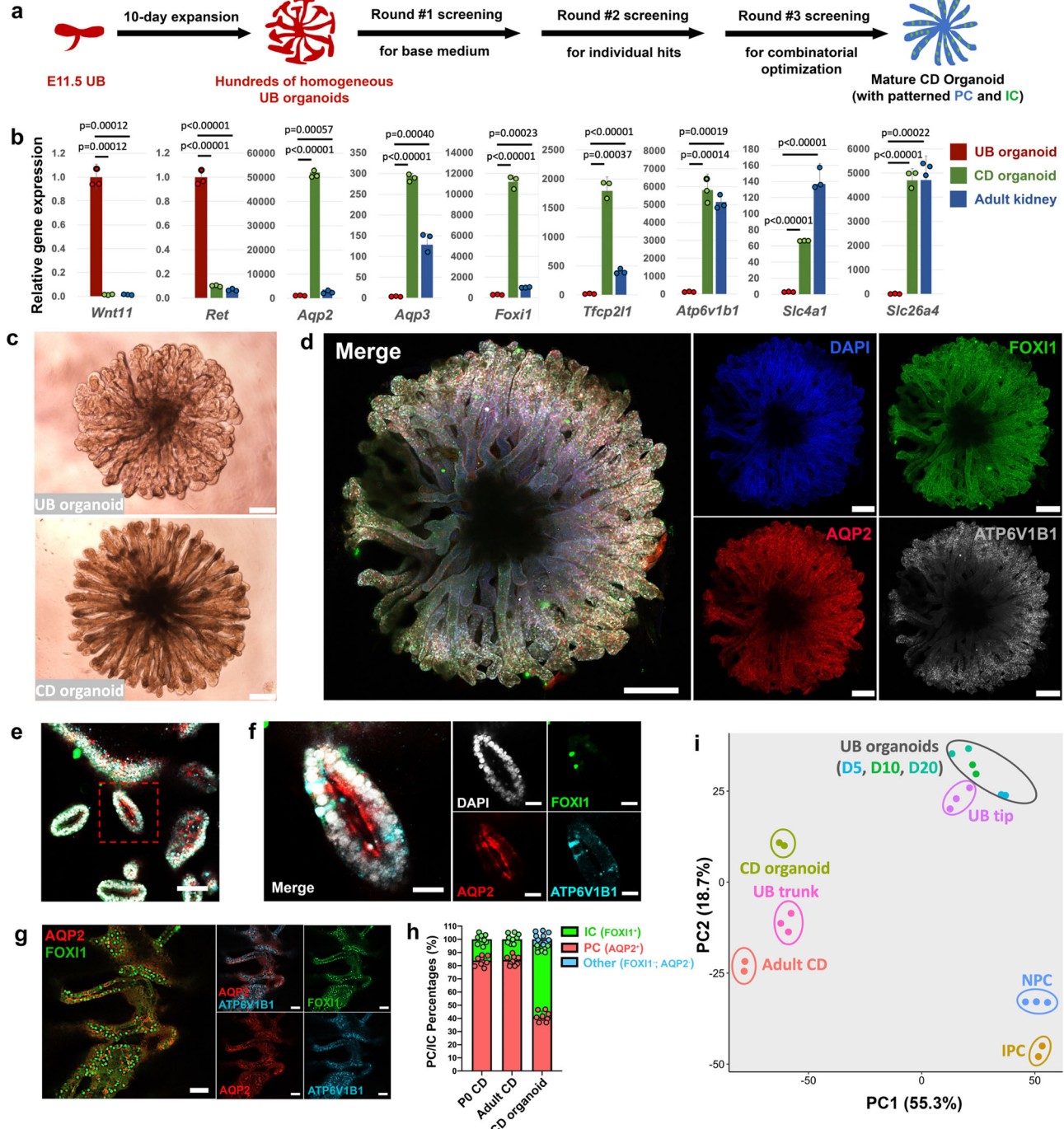

**Fig. 2 Generating mature and highly organized CD organoids from UB organoids. a** Schematic of the screening strategy for identifying differentiation culture condition to generate CD organoids from UB organoids. **b** qRT-PCR analyses of UB (red) and CD (green) organoids for UB progenitor markers *Wnt11* and *Ret*; PC markers *Aqp2* and *Aqp3*; IC markers *Foxi1*, *Atp6v1b1*, *Slc4a1*, and *Slc26a4*; and *Tfcp2l1* that is expressed in both PC and IC. Adult mouse kidney (blue) was used as control. The significance was determined by two-tailed unpaired Student's *t* tests. *n* = 3. **c** Bright field images showing the morphologic differences between UB and CD organoids. **d**–**g** Whole-mount immunostaining analyses of CD organoids for PC marker AQP2, and IC markers FOXI1 and ATP6V1B1, showing the distribution of these two cell types within the organoid. **f** Shows the higher-power images for the boxed region in (**e**). **h** Comparison of PC and IC ratios in postnatal Day 0 (P0) mouse CD, adult mouse CD, and CD organoids. Whole-mount immunostaining images (CD organoids), or section staining images (P0 and adult kidneys) stained for AQP2 (PC) and FOXI1 (IC) were quantified for ratios of PC and IC in the CD organoid or the kidney's collecting duct. Each column represents counts from nine different fields of view (*n* = 9, three fields were randomly selected from each of the three biologically independent samples). **i** Principal component analysis (PCA) of RNA-seq data. Different colors and oval circles represent different primary kidney cell populations (NPC, IPC, UB tip, UB trunk, and adult CD cells), or UB organoids cultured for 5 (D5), 10 (D10), and 20 days (D20), or CD organoids. Scale bars: **c**, **d** 200 μm, **e** 100 μm, **f** 25 μm, **g** 40 μm. All data are presented as mean ± s.d. Source data are provided as a Source Data file.

the commercially available APEL medium for sustaining kidney organoid generation[9]. Combinations of FGF9, EGF, and Y27632 were tested, together with the two different base media (Supplementary Fig. 3a). After 7 days of differentiation in the various conditions, we observed that the hBI + FGF9 + Y27632 condition enabled the survival of organoids and permitted spontaneous basal differentiation, as assayed by a modest induction of both PC (Aqp2) and IC (Foxi1) specific gene expression (Supplementary Fig. 3b, c).

To enhance the efficiency of differentiation, we carried out a 2nd round of screening identifying molecules that strongly induced the expression of Aqp2 and/or Foxi1 under the hBI + FGF9 + Y27632 condition. Agonists or antagonists targeting major developmental pathways (e.g., TGF-β, BMP, Wnt, FGF, Hedgehog, and Notch) were tested, together with hormonal inputs known to regulate PC or IC activity (aldosterone and vasopressin). BMP7, DAPT (a Notch pathway inhibitor), JAKI (JAK inhibitor I), and PD0325901 (MEK inhibitor) dramatically increased both Aqp2 and Foxi1 expression, while JAG-1[42] (Notch agonist) and aldosterone led to a preferential increase in Foxi1 expression, and vasopressin to enhanced Aqp2 expression (Supplementary Fig. 3d–f). In a 3rd round of screening, testing of various combinations of these factors led to the identification of an optimized CD differentiation medium (CDDM, Supplementary Table 8) supplemented with FGF9, Y27632, DAPT, PD0325901, aldosterone, and vasopressin.

**Generating mature and highly organized CD organoids from UB organoids.** Seven days of UB organoid culture in CDDM resulted in a morphologically elongated CD organoid phenotype (Fig. 2c). qRT-PCR revealed a marked decrease in the expression of the UPC genes (Wnt11 and Ret) and a concomitant elevation in the expression of PC-specific water transporter encoding genes (Aqp2 and Aqp3[40]) and IC-specific transcription factor (Foxi1), proton pump (Atp6v1b1[43]), and Cl⁻/HCO3⁻ exchangers (Slc4a1/Ae1[44], α-IC-specific; Slc26a4/Pendrin[45], β-IC-specific) (Fig. 2b). Immunostaining confirmed the presence of AQP2, AQP3, FOXI1, TFCP2L1[46], and ATP6V1B1 in the CD organoids (Fig. 2d–g and Supplementary Fig. 3g–j). Differentiating CD organoids displayed a clear lumen (Fig. 2e, f), and the organization of PC and IC cell types reflected that of the postnatal mouse kidney CD in which FOXI1⁺/ATP6V1B1⁺/TFCP2L1⁺/KIT⁺[47,48] ICs were dispersed in AQP2⁺/AQP3⁺ PCs (Supplementary Fig. 3k–n). Further, in some areas, AQP2 and AQP3 showed a differential subcellular localization, AQP2 to the apical luminal facing surface and AQP3 to basolateral plasma membrane, reflecting the normal cellular distribution of these critical components of water trafficking through PC cells (Fig. 2e, f and Supplementary Fig. 3i, j). PC and IC ratios in the CD organoids indicated a higher IC portion (50–55%) over PC portion (40–45%) than observed in the neonatal and adult kidney, a likely reflection of DAPT-mediated Notch inhibition in CDDM culture (Fig. 2h); in vivo, IC-derived Notch ligand signaling inhibits the IC fate[46,49–51].

To better define the identity of the CD organoids, we used RNA-seq to profile the transcriptome of the organoids. These data were compared with mouse CD (mCD) freshly isolated by FACS from the kidney of adult Hoxb7-Venus mice, as well as UB organoids, and prior RNA-seq data for primary UB tip and UB trunk populations[35], NPCs and IPCs[36]. PCA showed a clear separating of CD organoids from the immature UB tip and UB organoid populations, and similar grouping to UB trunk and primary mCD (Fig. 2i), supporting an expected cell maturation of CD organoids. Importantly, the CDDM differentiation protocol was highly reproducible when testing UB organoids derived from

different genetic backgrounds (Supplementary Fig. 3o). The evidence above support that we have generated a mature and patterned kidney organoid that recapitulates the adult kidney collecting system, in a chemically defined manner.

**Generating engineered kidney from expandable NPCs and UBs.** The availability of expandable NPCs[14,30] and UPCs provides the scalable building blocks required for making a kidney. As a proof-of-concept, we examined whether combining these cell types could generate a model mimicking key features of in vivo kidney development, such as reiterative ureteric branching and nephron induction, and morphogenesis and patterning of differentiating derivatives (Fig. 3a).

NPCs in our long-term culture model grow as 3D aggregates. To mimic the natural organization of NPCs capping UB tips in the kidney anlagen, we manually excavated a cavity in 3D cultured NPCs (expanded several billion-fold over for 6–12 months of culture) and inserted a cultured UB organoid tip. The engineered kidney structures were transferred onto an air–liquid interface to facilitate further kidney organogenesis. Over 7 days of culture, the inserted Hoxb7-Venus UB organoid tip underwent extensive branching (Fig. 3b) generating a tubular network extending from the center of the structure to the periphery. Further, NPCs generated nephron-like cell types including PODXL⁺/WT1⁺ podocytes and LTL⁺ proximal tubules (Fig. 3c).

To determine whether the engineered kidney also formed a connection between nephron and CD, we engineered kidneys comprising Hoxb7-Venus UB and wild-type NPCs. In this way, all progeny of the UB organoid could be tracked by Venus expression. Co-staining of the engineered kidney structure with CDH1 and GATA3-specific antibodies identified a clear fusion of CDH1⁺/Venus⁻ distal nephron with CDH1⁺/Venus⁺ CD. Importantly, GATA3 expression was strong in the entire Venus⁺ CD structure, but progressively dropped along the distal-to-proximal axis of the distal nephron, as observed in vivo[52,53] (Fig. 3d). Thus, the engineered kidney established a luminal interconnection between the nephron and CD, an essential morphological event for kidney function. Engineered kidney development was robust: ~80% of engineered kidneys underwent a similar developmental program, with most failures likely reflecting technical issues in manual construction (Fig. 3e). Taken together, engineered kidney with interconnected nephron and CD can be efficiently generated from expandable NPCs and UBs.

**Performing efficient gene editing in the expandable UB organoid.** The UB and CD models could provide an accessible in vitro complement to the mouse models for in-depth mechanistic studies and drug screening. Here, efficient gene overexpression (OE) or gene knockout (KO) would significantly extend the capability and utility of the in vitro model (Fig. 3f). As a proof-of concept, GFP OE and GFP KO UB organoids were generated. For GFP OE, we used a standard lentiviral system to introduce GFP under the control of a CMV promoter[54]. However, even at a very high titer, the lentiviral infection efficiency of the intact T-shaped UB was low. However, dissociating T-shaped UB or UB organoids into a single-cell suspension prior to infection dramatically improved the infection efficiency. Widespread GFP activity was observed in resulting UB organoids after reaggregation of infected cells (Fig. 3g). To test KO, we targeted GFP in Rosa26-Cas9/GFP UB organoid, in which Cas9 and GFP are constitutively expressed from the Rosa26 loci[38] (Supplementary Fig. 2e). A mix of three different lentiviral constructs, expressing three different single-guide RNAs (sgRNAs) with Cas9 targeting sites 100–150 bp

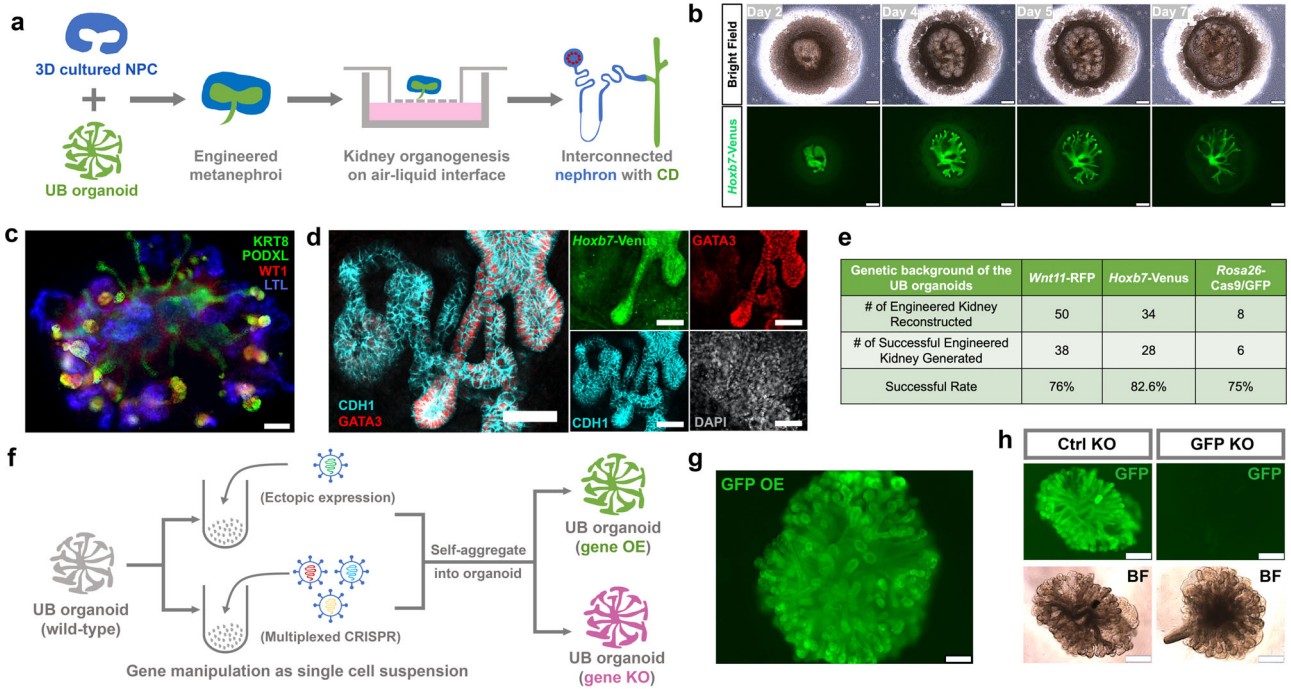

**Fig. 3 Generating engineered kidney from expandable NPCs and UBs and gene editing of the UB organoid. a** Schematic of the engineered kidney reconstruction and organotypic culture procedure. **b** Time course images (bright field and *Hoxb7*-Venus) showing the branching morphogenesis of the engineered kidney reconstructed at air–liquid interface at Days 2, 4, 5, and 7. Scale bars, 200 µm. **c** Immunostaining of the engineered kidney (Day 7) constructed from *Wnt11*-RFP UB organoid and wild-type NPCs for UB/CD marker KRT8, nephron marker PODXL and WT1 (podocytes) and LTL (proximal tubule). Note that both KRT8 and PODXL were stained green. The round structures that co-stain with WT1 are podocytes of the nephron. UB-derived structures do not co-stain with WT1. Scale bar, 100 µm. **d** Immunostaining of the engineered kidney constructed from *Hoxb7*-Venus UB organoid and wild-type NPC (Day 10) for GATA3 and CDH1. Scale bars, 50 µm. **e** Summary of engineered kidney generation experiments. **f** Schematic of gene overexpression and gene knockout procedures in the UB organoid. OE overexpression, KO knockout. **g** Fluorescence image of GFP overexpression (GFP OE) in wild-type UB organoid. Scale bar, 200 µm. **h** Knockout of GFP in *Rosa26*-Cas9/GFP UB organoid using multiplexed sgRNAs ("GFP KO," right panels) targeting the coding sequence of GFP. Multiplexed non-targeting sgRNAs were introduced to the organoid as control ("Ctrl KO," left panel). Note the gene-edited single cells self-organized into typical branching organoid morphology. Scale bars, 400 µm.

apart[55], gave a highly efficient, GFP sgRNA-specific, multiplexed CRISPR/Cas9 KO, demonstrating effective KO in UB organoid cultures (Fig. 3h).

**Generating human UB organoids from primary human UPCs (hUPCs).** The successful generation of mouse UB and CD organoids prompted us to test whether the system can also derive human UB and CD organoids. To achieve this, we developed a method to generate expandable human UB organoids from primary human UPCs (hUPCs) (Fig. 4a). Similar to their murine counterparts, hUPCs within UB tips, express *RET*[52,53,56] and *WNT11* (GUDMAP/RBK Resources, https://www.gudmap.org). Using an anti-RET antibody raised against the extracellular domain of RET that recognizes RET⁺ hUPCs (Fig. 4b), we performed FACS enrichment of RET⁺ cells from human kidneys between 9 and 13 weeks of gestational age and examined their growth in modified UBCM conditions. A robust human UBCM (hUBCM, Supplementary Table 7) culture condition was identified that sustained the long-term expansion (an estimated $10^8$–$10^9$ fold expansion over 70 days) as branching UB organoids (Fig. 4c). UPC marker gene expression was maintained in human UB cultures at level comparable to that of the human fetal kidney (Fig. 4d). Thus, we provide the proof-of-concept that expandable human UB organoid can be derived from purified RET⁺ primary human UPCs.

**Generating iUB and iCD organoids from hPSCs.** To determine whether UB and CD organoids could be generated from hPSC-

derived UPCs, we first genetically engineered H1 human embryonic stem cells (hESCs) with a knockin dual-reporter system where mCherry was expressed from the *PAX2* locus (*PAX2*-mCherry) and GFP from the *WNT11* locus (*WNT11*-GFP) (Supplementary Fig. 4a–e and Supplementary Methods). Using this reporter line, we first tested whether *PAX2*⁺/*WNT11*⁺ hUPCs can be generated following previously reported directed differentiation protocols that generated UB-like cells[23–26]. After directed differentiation, we confirmed the expression of *PAX2*-mCherry, but failed to observe the expression of *WNT11*-GFP, suggesting that the differentiation efficiency to generate hUPCs was relatively low following existing protocols. Relying on hUBCM's role in de novo hUPC induction and stabilization and a modified differentiation protocol, we were able to establish a stepwise protocol that resulted in high-quality hUPC cultures, which generated branching UB (iUB) organoids that underwent maturation to induced CD (iCD) organoids (Fig. 4e and Supplementary Table 9).

The UB is derived from the nephric duct (ND), which originates from primitive streak (mesendoderm (ME))-derived anterior intermediate mesoderm[2,4]. Consistent with this developmental trajectory, following a 7-day directed differentiation, we were able to first observe the expression of ME marker T on Day 3 of differentiation in most cells (Supplementary Fig. 4f), followed by the formation of large numbers of compact cell colonies that are GATA3⁺/SOX9⁺/PAX2⁺/PAX8⁺/KIT⁺/KRT8⁺ on Day 7 of differentiation, suggesting the generation of potential precursor cells of the UB lineage (Supplementary Fig. 4g). Consistent with

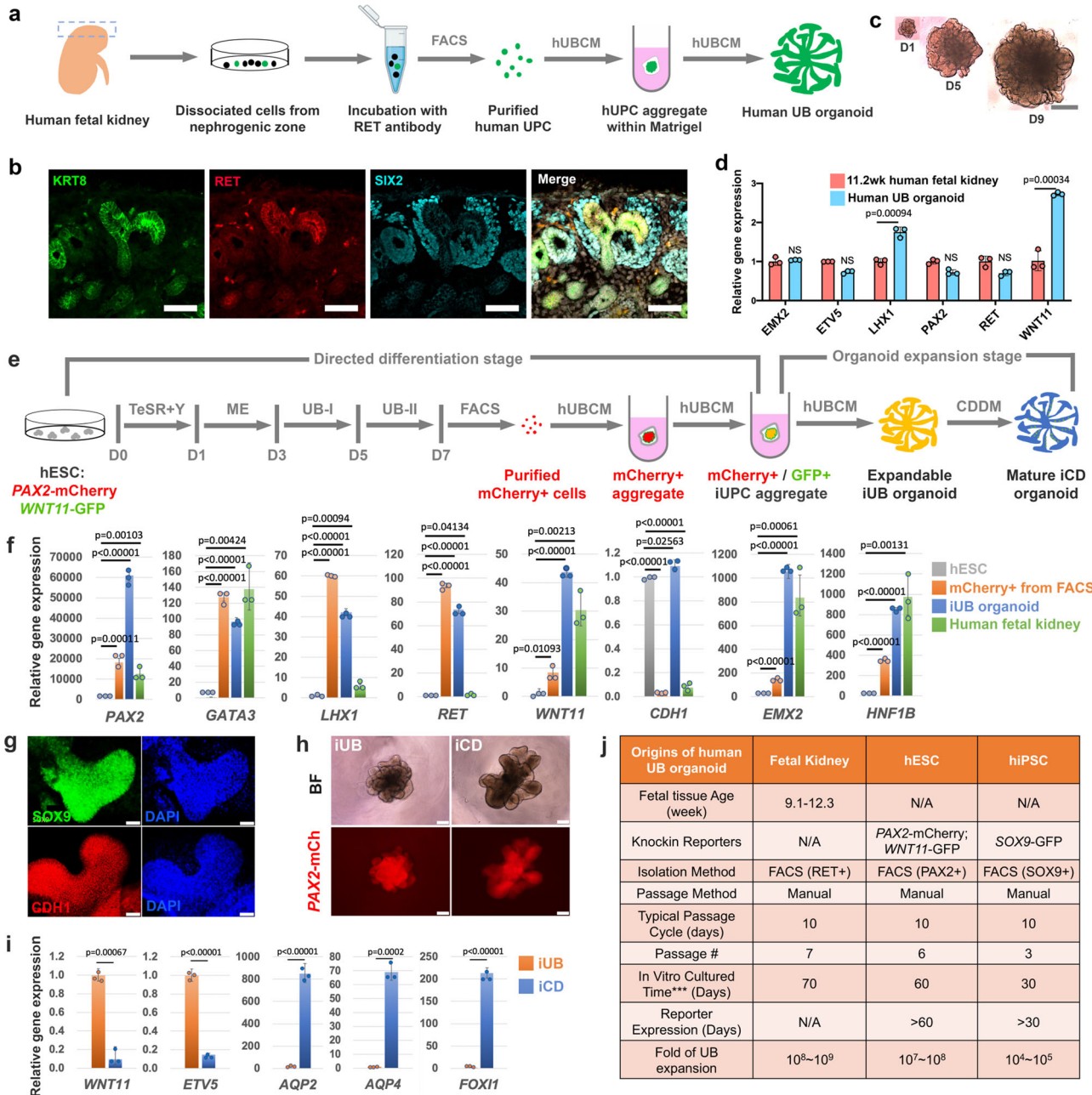

**Fig. 4 Generating human UB and CD organoids from primary human UPCs and dual-reporter human pluripotent stem cells. a** Schematic of the purification of primary human UPCs from the nephrogenic zone (illustrated as boxed region) of the human fetal kidney (9–13 weeks of gestational age) and the derivation of human UB organoid. **b** Immunostaining of the human fetal kidney nephrogenic zone for UB tip marker RET (red), broad UB lineage marker KRT8 (green), and NPC marker SIX2 (cyan). Scale bars, 50 μm. **c** Time course bright field images showing the growth of human UB organoid derived from primary human UPCs in a typical passage cycle at Days 1 (D1), 5 (D5), and 9 (D9). Scale bar, 200 μm. All images in (**c**) have been scaled to share the same scale bar with the D9 image. **d** qRT-PCR analyses of human UB organoid (cultured for 54 days) derived from primary human UPCs for various UB markers as indicated. Human fetal kidney from 11.2-week (11.2 weeks) gestational age was used as control. **e** Schematic of the stepwise differentiation from *WNT11*-GFP/*PAX2*-mCherry dual-reporter hESC line into iUB and iCD organoid. (TeSR mTeSR1 medium, Y Y27632, ME mesendoderm stage medium, UB-I UB Stage I medium, UB-II UB Stage II medium). **f** qRT-PCR analyses of the FACS purified mCherry$^+$ cells (orange) and the iUB organoid (blue, cultured for 50 days) for various UB markers as indicated. Undifferentiated H1 hESCs (gray) and human fetal kidney (green, 11.2-week gestational age) were used as controls. **g** Whole-mount immunostaining of the expandable iUB organoid for UB markers SOX9 and CDH1. Scale bars, 50 μm. **h** Bright field (BF) and *PAX2*-mCherry (*PAX2*-mCh) images of expandable iUB organoid (left panels) and mature iCD organoid (right panels). Scale bars, 200 μm. **i** qRT-PCR analyses of the iUB organoid (orange, cultured for 49 days) and iUB-derived iCD organoid (blue), for UB (*WNT11*, *ETV5*), PC (*AQP2*, *AQP4*), and IC markers (*FOXI1*). **j** Summary of human UB organoid derivation from different sources and their expansion in vitro. ***, this is the culture time we achieved before our lab shutdown due to coronavirus outbreak. The maximum organoid culture time and expansion could be much greater. All data are presented as mean ± s.d. In **d**, **f**, **i**, the significance was determined by two-tailed unpaired Student's *t* tests; NS not significant; *n* = 3. Source data are provided as a Source Data file.

the immunostaining results, we were able to identify a *PAX2*-mCherry[+] population (13.1%) by FACS on Day 7. However, at this stage, the *PAX2*-mCherry[+]/*WNT11*-GFP[+] population was very rare (0.4%), preventing further characterization or culture (Supplementary Fig. 4h and 8b). However, further culture of *PAX2*-mCherry+ cells in the 3D/hUBCM culture conditions activated *WNT11*-GFP reporter expression at around 3 weeks, and the structure started to show a branching morphology (Supplementary Fig. 4i). We refer to the *PAX2*-mCherry[+]/*WNT11*-GFP[+] branching structure an "iUB" organoid hereafter. Importantly, these iUB organoids could be expanded stably in 3D/hUBCM for at least 2 months without losing reporter gene expression (Fig. 4j). Consistently, qRT-PCR analysis confirmed that *WNT11* expression was low in the mCherry[+] cells purified from FACS, but was dramatically elevated in the iUB organoid. Furthermore, even though UB marker genes *PAX2, GATA3, LHX1,* and *RET* were greatly elevated on Day 7 of differentiation, while *WNT11, CDH1, EMX2,* and *HNF1B,* showed comparable levels of expression to the human fetal kidney only after extended hUBCM culture, suggesting that hUBCM promoted transition from a common ND to a specific ureteric epithelial precursor (Fig. 4f). In addition to qRT-PCR, expression of marker genes *SOX9* and *CDH1* in the iUB organoid was detected at the protein level by immunostaining (Fig. 4g), further confirming the identity of the iUB organoid.

To determine whether the expandable iUB organoid retained the potential to generate an iCD organoid after long-term expansion. iUB organoids were subjected to differentiation with the CDDM medium identified for mouse UB-to-CD transition. After 14 days of differentiation in CDDM, the human iUB organoid grew and elongated, with maintained *PAX2*-mCherry expression (Fig. 4h). More importantly, the expression of UPC markers *WNT11* and *RET* was greatly diminished, while CD marker genes *AQP2, AQP4,* and *FOXI1* were dramatically elevated, suggesting the successful transition from iUB to iCD (Fig. 4i).

To test whether expandable iUB organoids could be generated from human-induced pluripotent stem cells (hiPSCs), we employed *SOX9*-GFP hiPSC[57] for differentiation and purified the *SOX9*-GFP[+] UB precursor cells on Day 7 of differentiation (Supplementary Figs. 4j and 8c). Similar to hESC-derived iUBs, following an extended culture in hUBCM, we were able to derive *SOX9*-GFP iUB organoids that expanded stably with retained *SOX9*-GFP expression throughout (Fig. 4j and Supplementary Fig. 4k–m). Taken together, these results support the conclusion that expandable iUBs and mature iCDs organoid can be derived from hESC and hiPSC lines.

**Generating iUB and iCD organoids independent of reporter hPSC lines**. The reporter hPSC lines are useful in developing iUB differentiation protocols, but if iUB organoids can only be derived from these reporter hPSCs, its applications will be significantly limited. To solve this problem, we next developed a method to derive iUB organoid from any given hPSC line in the absence of reporter (Fig. 5a). In this method, after 7 days of differentiation, sorting of KIT[+] cells was used to enrich the precursor population, rather than sorting based on *PAX2*-mCherry or *SOX9*-GFP reporters. With further refinement of our stepwise iUB differentiation protocol and a refined hUBCM (hUBCM-v2, Supplementary Tables 7 and 9), long-term expandable branching iUB organoids can be derived within 12 days from hPSCs with high efficiency. Following this protocol, as proof-of-concept, we have successfully derived iUB organoids from three different hPSC lines, including two hESC lines (Fig. 5 and Supplementary Fig. 5) and one hiPSC line (Supplementary Fig. 6).

KIT was previously reported as a surface marker that can be used to enrich UB-like cells upon hPSC differentiation[23]. Interestingly, we noticed that KIT[+] cells are frequently co-stained with PAX2 (Supplementary Fig. 4g). We thus hypothesized that FACS of KIT[+] cells will enrich for PAX2[+] precursor cells similar to the sorting of *PAX2*-mCherry[+] cells using our *PAX2*-mCherry reporter line. Starting from our *WNT11*-GFP/*PAX2*-mCherry hESC line, after 7 days of differentiation, 36.1% of the cells were KIT[+] (Supplementary Fig. 5a and 8d). Further culture of these KIT[+] cells in hUBCM-v2 showed a much faster induction of *WNT11*-GFP expression than using hUBCM (5–7 days with hUBCM-v2, as shown in Supplementary Fig. 5b, vs. ~3 weeks with hUBCM as shown in Supplementary Fig. 4i). Importantly, accompanying the expression of *WNT11*-GFP, the organoid started to show the typical branching morphology (Supplementary Fig. 5b), and can since be stably passaged and expanded billions of folds either manually or as single cells for at least 70 days by the time our manuscript is submitted (Supplementary Fig. 5c, h).

To determine whether gene expression in the iUB organoid is also stably maintained over long-term culture, we collected iUB organoids 33, 49, and 66 days after the initiation of culture in hUBCM-v2, and compared their gene expression by qRT-PCR with undifferentiated hPSCs, KIT[+] precursor cells, and human fetal kidney tissue. Consistent with our previous finding with sorted *PAX2*-mCherry[+] precursors (Fig. 4f), the sorted KIT[+] precursor cells also showed strong expression of *PAX2, GATA3, LHX1,* and *RET,* but the expression of *WNT11, CDH1, EMX2,* and *HNF1B* was only induced after further programming in the presence of hUBCM-v2 (Fig. 5b). Importantly, gene expression of all these markers in the iUB organoids were maintained stably throughout the culture period, at levels comparable or higher than that of human fetal kidney tissue, indicating the robustness of our method. Whole-mount immunostaining (Fig. 5c–e) or section staining (Supplementary Fig. 5d–f) of the iUB organoids for various general UB lineage markers (KRT8, PAX2, PAX8, GATA3, and CDH1) or UB tip markers (RET and SOX9) further confirmed the expression of all these genes at the protein level. Importantly, quantification of the immunostaining results indicated that more than 95% of cells in the organoids showed homogeneous expression of all markers (Fig. 5f), suggesting that the majority of the cells in the iUB organoid are of the UB progenitor identity.

To determine whether the iUB organoid can generate iCD organoid, we further developed a refined CDDM (human CD differentiation medium (hCDDM), Supplementary Table 8), which efficiently induced the mRNA expression of various PC (*AQP2, AQP3,* and *AQP4*) and IC (*FOXI1*) markers hundreds to thousands of fold within 14 days of differentiation from iUB organoids, accompanying the reduction of UB tip genes *WNT11* and *RET* (Fig. 5g and Supplementary Fig. 5g). Given the limited availability of validated antibodies, we were only able to examine AQP3 and FOXI1 at the protein level (Supplementary Fig. 7g–m). Approximately, 20–30% of iCD organoid cells were AQP3[+] but we were not able to observe FOXI1[+] ICs (Fig. 5h). These results suggest that long-term expandable iUB organoids generate PC and IC-like cells but cells generated in hCDDM do not attain a fully mature CD cell fates.

Importantly, similar iUB organoids were also generated from a second hESC line (H1) and from the *SOX9*-GFP reporter hiPSC line. After 7 days of differentiation, 55.4% of H1 cells and 43.9% of *SOX9*-GFP hiPSCs were identified as KIT[+] on FACS (Fig. 5i and Supplementary Figs. 6a, 8a, e). Further culturing of these KIT[+] cells in hUBCM-v2 derived iUB that can be stably expanded as branching iUB organoids (Fig. 5j and Supplementary Fig. 6b, c). qRT-PCR further confirmed that gene expression in

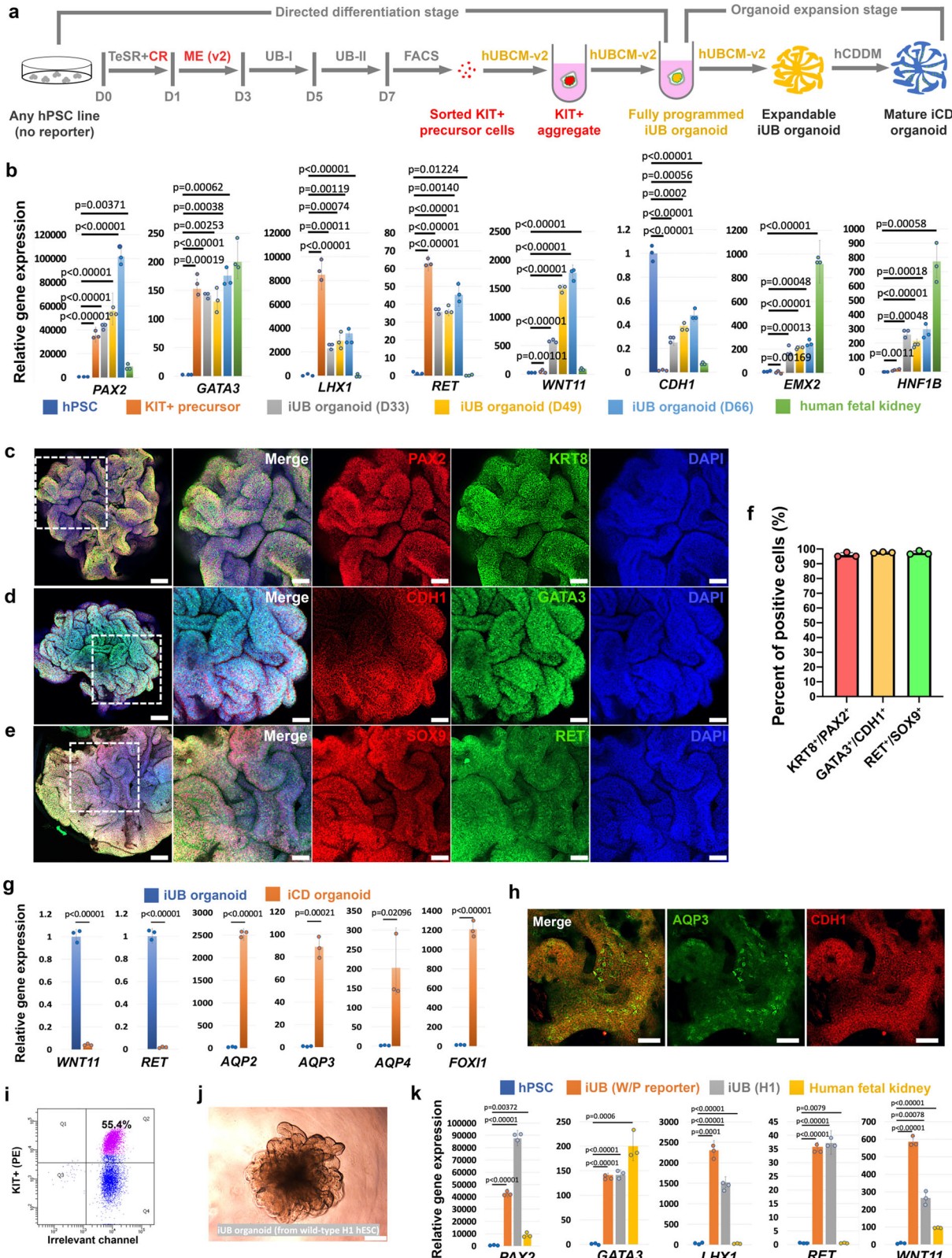

the H1 hESC-derived iUB organoid is similar to the iUB organoid derived from our dual-reporter hESC line shown above (Fig. 5k). Similarly, qRT-PCR (Supplementary Fig. 6d), whole-mount immunostaining (Supplementary Fig. 6e–h), and section staining (Supplementary Fig. 6i–l) confirmed that the majority of the cells in the hiPSC-derived iUB organoids are of the UB progenitor identity.

**Modeling kidney development and disease using mouse and human UB organoids.** GDNF is a critical signal in both mouse and human UB culture. In vivo, GDNF secreted by MM cells surrounding UPC-containing branch tips signals via RET, with its co-receptor GFRA1, to maintain the UPC state and stimulate UB branching morphogenesis[2,4]. Loss of the activity of these genes results in a CAKUT syndome[2,4,5,58,59]. We employed

**Fig. 5 Generating human iUB and iCD organoids from human pluripotent stem cells independent of reporters. a** Schematic of the stepwise differentiation from any hPSC line into iUB and iCD organoid without using reporter. (TeSR mTeSR1 medium, CR CloneR, ME (v2) mesendoderm stage medium version 2, UB-I UB Stage I medium, UB-II UB Stage II medium). **b–h** Characterizations of iUB or iCD organoids derived from the *WNT11*-GFP/*PAX2*-mCherry hESC line independent of its reporters. **b** qRT-PCR analyses of the FACS purified KIT$^+$ precursor (orange), KIT$^+$ precursor-derived iUB organoids cultured for 33 (D33, gray), 49 (D49, yellow), and 66 days (D66, light blue) for various UB markers as indicated. Undifferentiated H1 hESCs (dark blue) and human fetal kidney (green, 11.2-week gestational age) were used as controls. **c–e** Whole-mount immunostaining of the expandable iUB organoid for various UB markers as indicated. The four panels on the right represent the boxed region in the left panel. Scale bars, left panel, 100 µm; right four panels, 40 µm. **f** Quantification of percentages of iUB cells stained positive for different UB markers in Fig. 5c–e. Each column represents counts from three different fields of view ($n = 3$). **g** qRT-PCR analyses of the iUB organoid (blue) and iUB-derived iCD organoid (orange), for UB (*WNT11*, *RET*), PC (*AQP2*, *AQP3*, *AQP4*), and IC markers (*FOXI1*). **h** Whole-mount immunostaining of the iUB-derived iCD organoid for PC marker AQP3 and broad CD marker CDH1. Scale bars, 50 µm. **i** Flow cytometry analysis of KIT$^+$ precursor cells differentiated from wild-type H1 hESC. **j** Bright field image of a typical iUB derived from wild-type H1 hESC. Scale bar, 250 µm. **k** qRT-PCR analyses of the iUB organoids derived from the dual-reporter hESC line (W/P reporter, orange, cultured for 33 days) or wild-type H1 hESC line (H1, gray, cultured for 30 days) for various UB markers. Undifferentiated H1 hESCs (blue) and human fetal kidney (yellow, 11.2-week gestational age) were used as controls. All data are presented as mean ± s.d. In **b**, **g**, **k**, the significance was determined by two-tailed unpaired Student's *t* tests; $n = 3$. Source data are provided as a Source Data file.

CRISPR/Cas9 system to KO *Ret/RET* in mouse and human UB organoids predicting as in vivo, UB organoid development in vitro would be *Ret/RET*-dependent (Fig. 6a). UB organoids were infected with lentivirus expressing Cas9 and two independent sgRNAs targeting *Ret/RET* (in lentiCRISPR-v2 vector), while a control group received sgRNA without Cas9 (in lentiGuide-puro vector). As expected, the control mouse UB organoids grew normally with maintained branching morphogenesis upon lentiviral infection and puromycin selection, while both *Ret* KO UB organoids stopped branching (Fig. 6b). Whole-mount immunostaining of control and *Ret* KO UB organoids confirmed a dramatic reduction (more than 95%) of RET expression 6 days after lentiviral infection in the *Ret* KO UB organoids, demonstrating the successful removal of *Ret* (Fig. 6c, d). Consistent with the defect in branching morphogenesis in the *Ret* KO organoids, genes enriched in UB tip, *Wnt11*, and *Lhx1*[35] were reduced dramatically, and the expression of common UB lineage markers *Pax2* and *Gata3* were also decreased (Fig. 6e). KO of *RET* in the human iUB organoid also resulted in the arrest of branching morphogenesis in both of the *RET* KO organoids receiving two different sgRNAs (Fig. 6f) and a loss of *RET* immunostaining in more than 95% of cells with both sgRNAs (Fig. 6g, h and Supplementary Fig. 7a–f). Interestingly, even though *WNT11* and *GFRA1* expression was significantly reduced in both *RET* KO iUB organoids, the expression of *LHX*, *GATA3*, *PAX2*, as well as *ETV5* and *SOX9*, did not show consistent changes in both *RET* KO organoids (Fig. 6i). These results suggest that species-specific regulatory network downstream of *Ret/RET* might govern UB progenitor fate, consistent with previous observations of convergent and divergent mechanisms of nephrogenesis between mouse and human[36,52]. Taken together, we provide a proof-of-concept for recapitulating kidney development and disease using mouse and human UB organoid models.

## Discussion

In this study, we report 3D culture models enabling the expansion and differentiation of mouse and human UPCs. The organoid culture medium effectively replaces cell interactions within the nephrogenic niche of the developing mammalian kidney, with a chemically defined synthetic niche capable of maintaining UPC identity. Consistent with mouse genetics studies, signaling pathways that play key roles in kidney branching morphogenesis, such as GDNF[32,60,61], FGF[62], RA[63–65], and Wnt[66,67] signaling, are also essential in maintaining UPC identity in UB organoids. UPC cloning efficiency is not very high in the UBCM culture. Similar to our 3D NPC culture[14,30], it is likely that cell–cell contact is important for maintaining the best tip identity, as aggregated UPCs, or manually passaged UB organoids as small

cell clusters, can maintain *Wnt11*-RFP homogeneously. Better understanding of cell–cell contact and/or potential additional paracrine signals might help further improve the culture, thus allowing the development of a more robust clonal expansion method.

Leveraging our ability to produce large quantities of high-quality UPCs in the format of expandable branching UB organoids, we performed a screening that identified CDDM—a cocktail of growth factors, small molecules, and hormones that together can differentiate UB organoids into CD organoids with spatially patterned mature PCs and ICs. The molecular mechanisms underlying the UB-to-CD transition are still largely unknown. The in vitro organoid system provides a tool to study this process, and the chemically defined components in CDDM shed light on potential signals that trigger CD maturation in vivo. Despite the general difficulty of maturing stem-cell-derived tissues, our study shows that it is possible to achieve proper patterning and maturation in vitro, similar to what we observe in vivo, when starting from high-quality progenitor cells under appropriate culture conditions. The limited number of available antibodies precludes a more comprehensive characterization of the mature human PC and IC state. The current platform is a strong base for future studies to further improve CD cell maturation and assess physiological activity of differentiated cell types.

An interesting observation during the de novo human UB directed differentiation process was the induction of hUPC fate by hUBCM or hUBCM-v2. One possibility to explain this phenomenon is that these media could stabilize the rare and transient hUPC population generated from directed differentiation. Another possibility is that these media could promote cell fate transition from earlier WD-staged precursor cells to the hUPC fate. In support of both possibilities, our NPC culture medium, NPSR, has recently been reported to facilitate the generation of NPC-like cells in both directed differentiation[68] and transdifferentiation[69] settings. Future studies are warranted to understand how hUBCM and hUBCM-v2 contributes to hUPC fate specification.

The generation of an engineered kidney from expandable NPCs and UPCs provides a proof-of-concept for rebuilding a kidney in vitro from kidney-specific progenitor cells. The availability of expandable NPCs and UPCs provides the scalable building blocks required for making a kidney. The interaction between NPCs and UPCs is faithfully recapitulated, leading to the autonomous differentiation into interconnected nephron and CD structures. Interestingly, different from prior study[23], in our engineered kidney system, IPCs appear to be dispensable in reconstructing a branching kidney structure in vitro. It is likely

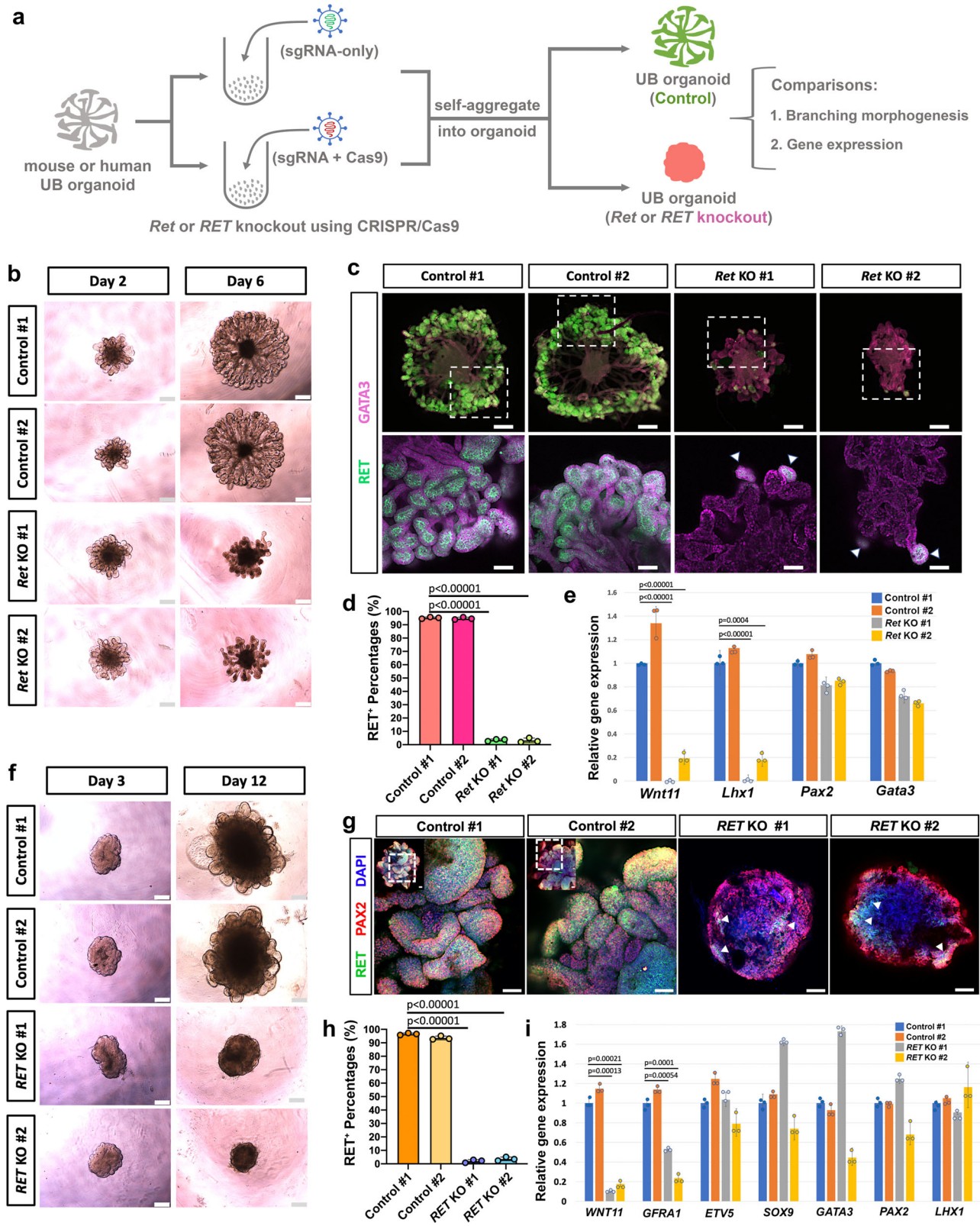

that one of our engineered kidney culture medium components, TTNPB, a small molecule analog of RA, substitutes RA production by IPCs, an essential mechanism for proper UB branching and kidney development in vivo[63–65]. Future efforts will require the integration of vascular progenitor cells, and a more in-depth evaluation of IPCs, to develop engineered structures for testing in animal models of organ transplantation.

Efficient genome editing in UB organoids opens up many applications using the UB and CD organoid platform. UB and CD organoids can be generated from available transgenic mouse strains that bear genetic mutations related to kidney development and disease. In addition, disease-relevant mutations can be introduced into the UB organoid directly, enabling the investigation of pathophysiology throughout the entire course of kidney

**Fig. 6 Modeling kidney development and disease using mouse and human UB organoids. a** Schematic of *Ret/RET* gene knockout procedures in the mouse or human UB organoid. **b** Bright field images showing the branching morphogenesis of mouse UB organoids 2 days (Day 2) and 6 days (Day 6) after lentiviral infection. Scale bars, 200 μm. **c** Whole-mount immunostaining of the control or *Ret* KO mouse UB organoids for UB markers GATA3 and RET after 6 days of lentiviral infection. Arrow heads indicate the few RET+ cells in the *Ret* KO organoids. Lower panels represent the boxed region in the upper panels. Scale bars, upper panels, 200 μm; lower panels, 40 μm. **d** Quantification of percentages of cells stained positive for RET in Fig. 6c. Each column represents counts from three different fields of view (n = 3). **e** qRT-PCR analyses of the control mouse UB organoids (blue and orange) and *Ret* KO mouse UB organoids (gray and yellow) for various UB markers 6 days after lentiviral infection. **f** Bright field images showing the branching morphogenesis of human iUB organoids 3 (Day 3) and 12 days (Day 12) after lentiviral infection. Scale bars, 200 μm. **g** Whole-mount immunostaining of the control or *RET* KO human iUB organoids for UB markers PAX2 and RET 12 days after lentiviral infection. Arrow heads indicate the few RET+ cells in the *RET* KO organoids. Scale bars, 40 μm. See also Supplementary Fig. 7a–f for images of individual fluorescent channels. **h** Quantification of percentages of human iUB cells stained positive for RET in Fig. 6g. Each column represents counts from three different fields of view (n = 3). **i** qRT-PCR analyses of the control human iUB organoids (blue and orange) and *RET* KO iUB organoids (gray and yellow) for various UB markers as indicated 12 days after lentiviral infection. All data are presented as mean ± s.d. In **d**, **e**, **h**, **i**, the significance was determined by two-tailed unpaired Student's *t* tests; n = 3. Source data are provided as a Source Data file.

branching morphogenesis, from the UB branching period to the mature CD stage. Our proof-of-concept *Ret/RET* KO experiment demonstrated that our UB organoid system can recapitulate genetic malfunction of branching morphogenesis in vitro as that of in vivo. Our results also shed light on the potential different regulatory mechanisms downstream of *Ret/RET* in mouse and human. Our system offers a unique platform to further investigate how human *RET* mutations identified in the human CAKUT patients might contribute to congenital kidney malformation. The ability to produce large quantities of UB and CD organoids also provides a platform for drug screening. The mature PCs and ICs present in the CD organoids are potential sources for cell replacement therapies for patients with CD damage. In conclusion, the UB and CD organoid system provides a powerful tool for studying kidney development, modeling kidney disease, discovering drugs and, ultimately, regenerating the kidney.

## Methods

**Human tissues.** All human fetal kidney samples were collected under Institutional Review Board approval (USC-HS-13-0399 and CHLA-14-2211). Following the patient decision for pregnancy termination, the patient was offered the option of donation of the products of conception for research purposes, and those that agreed signed an informed consent form. No financial gain arose on donation of tissue. This did not alter the choice of termination procedure, and the products of conception from those that declined participation were disposed of in a standard fashion. The only information collected was gestational age and whether there were any known genetic or structural abnormalities.

**Mice.** All animal work was performed under Institutional Animal Care and Use Committee approval (USC IACUC Protocol # 20829). Swiss Webster mice were purchased from Taconic Biosciences (Model # SW-F, MPF 4 weeks). *Sox9*-GFP mice were kindly shared from Dr Haruhiko Akiyama[37]. *Wnt11*-RFP mice (JAX # 018683), *Hoxb7*-Venus mice (JAX # 016252), and *Rosa26*-Cas9/GFP (JAX #026179) were obtained from the Jackson Laboratory.

**hPSC lines.** Experiments using hPSCs were approved by the Stem Cell Oversight Committee (SCRO) of University of Southern California under protocol # 2018-2. hPSCs are routinely cultured in mTeSR1 medium in monolayer culture format coated with Matrigel and passaged using dispase.

**3D cultured NPC lines.** The 3D cultured NPC lines we used in this study were derived from E11.5 whole kidney cells of the wild-type Swiss Webster mouse strain, using an improved method we developed that can derive NPC lines from any mouse strain without the need for prior purification of NPCs[30]. These NPCs had been cultured 6–12 months (billions of billion-fold of expansion) before used for reconstruction of engineered kidney with UB organoids.

### Derivation of mouse UB organoid

*From intact T-shaped UB.* Male mice with the desired genotype (*Wnt11*-RFP, *Hoxb7*-Venus, *Sox9*-GFP, or *Rosa26*-Cas9/GFP) were mated with female Swiss Webster mice. Plugs were checked the next morning; midday of plug positive was designated as embryonic Day 0.5 (E0.5). Timed pregnant mice were euthanized at E11.5. Kidneys were dissected out from embryos using standard dissection techniques and transferred into a 1.5-mL Eppendorf tube on ice. Next, at least 500-μL

fresh, pre-warmed 0.1% (w/v) collagenase IV (Thermo Fisher, Cat. No. 17104019) was added into the tube and incubated at 37 °C for 20 min. After incubation, collagenase was removed and at least 500 μL of 10% FBS (1X DMEM, 1X Gluta-MAX-I, 1X MEM NEAA, 0.1 μM 2-Mercaptoethanol, 1X Pen Strep, 10% FBS) was added to resuspend the kidneys. One to three kidneys were transferred each time with 80–100-μL medium onto a 100-mm petri dish lid as a working droplet. UBs were isolated from the surrounding MM and other tissues using sterile needles (BD, Cat. No. BD305106) without damaging UBs. The isolated UBs can be temporarily left in the medium at room temperature for <30 min while dissecting other UBs. After all UBs were isolated, each UB was transferred together with 1–3-μL medium into an 8-μL cold Matrigel droplet at the bottom of one well of a U-bottom 96-well low-attachment plate, by using a P10 micropipette. The UB and Matrigel were mixed by pipetting gently 2–3 times. After all UBs were embedded in Matrigel, the plate was incubated at 37 °C for 20 min for the Matrigel to solidify. Then, 100 μL of mouse UBCM (mUBCM) was slowly added into each well and the plate was then transferred into an incubator set at 37 °C with 5% $CO_2$.

*From dissociated UB single cells.* For deriving UB organoid from dissociated UB single cells (e.g., for gene editing purpose), after the isolation of E11.5 T-shaped UBs from kidneys following the procedures described above, all UBs were collected into a 1.5-mL Eppendorf tube with the medium removed as much as possible. An appropriate amount (e.g., for 20 UBs, we use 200 μl, adjust accordingly) of pre-warmed Accumax cell dissociation solution (Innovative Cell Technologies, # AM105) was added into the tube, and the tube was then incubated at 37 °C for 20 min and gently tapped every 7–10 min. Then, an equal amount of 10% FBS was added into the tube to neutralize the Accumax and the mixture was pipetted gently 8–10 times to dissociate the UB into single cells. The tube was then centrifuged at $300 \times g$ for 5 min. After centrifugation, the supernatant was carefully removed and UB cells were resuspended in an appropriate amount of mUBCM (Y27632 was supplemented at 10-μM final concentration for the first 24 h) by pipetting gently 6–8 times. Cell density was measured using automatic cell counter (Bio-Rad, TC20). Approximately, 2000 cells were transferred into each well of a U-bottom 96-well low-attachment plate and extra amount of mUBCM (with 10-μM Y27632) was added to the well to make the final volume 100 μl per well. The plate was then centrifuged at $300 \times g$ for 3 min and transferred and cultured in a 37 °C incubator. After 24 h, the ~2000 single cells formed an aggregate autonomously and the aggregate was then transferred together with 1–3-μl medium into an 8-μl cold Matrigel droplet in another well of the U-bottom 96-well low-attachment plate using a P10 micropipette. The aggregate was pipetted gently 2–3 times to mix with Matrigel. After all aggregates were embedded in Matrigel, the plate was incubated at 37 °C for 20 min for the Matrigel to solidify. Last, 100 μl of UBCM was added slowly into each well and the plate was then transferred into an incubator set at 37 °C with 5% $CO_2$.

**Mouse UB organoid expansion and passaging.** Mouse UBCM was renewed with fresh medium every 2 days, and UB organoid was passaged every 5 days.

*Manual passaging as small tips.* UB organoid (with Matrigel) was first transferred from U-bottom 96-well plate onto a 100-mm petri dish lid with 80–100-μL medium using a P1000 pipette with the tip cut 0.5–1 cm to widen the diameter. Most of the Matrigel surrounding the organoid was removed using sterile needles under a dissecting microscope. A small piece of the organoid with 3–5 branching tips was cut using needles and then re-embedded into Matrigel droplet in a U-bottom 96-well low-attachment plate well and cultured in a 37 °C incubator following the same embedding procedure described above.

*Passaging as single cells.* UB organoid (with Matrigel) was first transferred from U-bottom 96-well plate onto a 100-mm petri dish lid with 80–100-μL medium using a P1000 pipette with the tip cut 0.5–1 cm to widen the diameter. Most of the Matrigel

surrounding the organoid was removed using sterile needles under a dissecting microscope. Organoid was then cut into small pieces using sterile needles (the smaller the piece, the easier to dissociate). All the pieces were transferred into 1.7-mL Eppendorf tubes (1–3 organoids per tube) with as little medium as possible using a P200 pipette, extra medium was removed from the tube. Two hundred to four hundred microliters of pre-warmed Accumax cell dissociation solution was added into the tube. Then followed the same procedure described above (in "Derivation of mouse UB organoid"—"From dissociated UB single cells," following the addition of Accumax) to dissociate organoid pieces into single cells and reaggregate for continuing culture.

**Derivation of human UB organoid**

*Organoid derivation from RET+ primary UPCs purified from human fetal kidney.* The kidney nephrogenic zone was dissected manually from each of fresh 9–13-week human fetal kidney, chopped into small pieces with surgical blade, and divided into 4–6 1.5-mL Eppendorf tubes. Tissues were washed with PBS and resuspended with 500 μL of pre-warmed Accumax per tube and the tubes were incubated at 37 °C with shaking for 25 min. Five hundred microliters 10% FBS was then added to neutralize the Accumax, and the mixture was pipetted ~25 times to dissociate the tissues into single cells. The mixture medium with kidney cells were then pooled together and sieved through a 40-μm cell strainer, then transferred into 1.5-mL Eppendorf tubes, centrifuged at $300 \times g$ for 5 min and the supernatant was carefully removed. All cell pellets from this preparation were then resuspended and combined into 300–400-μL cold FACS medium (1x PBS, 1X Pen Strep, 2% FBS) supplemented with a human anti-RET antibody at 1:200 dilution into one tube and incubated for 30 min on ice. The tube was gently tapped every 10 min to ensure mixing. After 30 min, 1-mL cold FACS medium was added into the tube. The tube was then centrifuged at $300 \times g$ for 5 min and the supernatant was carefully removed. Cell pellet was resuspended again in 500-μL cold FACS medium plus secondary antibody (Donkey anti-Goat, Alexa Fluor 568, Invitrogen, Cat. # A-11057) at 1:1000 dilution and incubated for 30 min on ice, with gentle mixing every 10 min. After the incubation, 1-mL cold FACS medium was added into the tube. The tube was then centrifuged at $300 \times g$ for 5 min and the supernatant was carefully removed. The pelleted cells were resuspended with 300–500-μL cold FACS medium plus DAPI at 1:2000 ratio, placed through 40-μm cell strainer and transferred into a FACS tube on ice before FACS. RET+ (Alexa Fluro 568) UPCs were then sorted out by FACS. The RET+ cells were collected in a 1.5-mL Eppendorf tube with 500-μL 10% FBS. The tube was centrifuged at $300 \times g$ for 5 min and the supernatant was carefully removed. Cell pellet was then resuspended in an appropriate amount of hUBCM (with the addition of Y27632 at 10-μM final concentration) and cell density was measured by automatic cell counter. ~2000–20,000 cells were transferred into each well of a U-bottom 96-well low-attachment plate and an appropriate amount of hUBCM (with 10-μM Y27632 for the first 24 h) was added to the well to make the final volume of 100 μL per well. After 24 h, UB cell aggregate was formed and embedded into an 8-μL cold Matrigel droplet in another well of the U-bottom 96-well low-attachment plate and cultured in a 37 °C incubator following the same embedding procedure described above (with hUBCM). After ~10–15 days of culture, epithelial tip structures could be seen budding out from the aggregate. These tip structures were dissected out and re-embedded into Matrigel and expanded as human UB organoid.

*Organoid derivation from human ESCs and iPSCs.* The hPSCs were pre-treated with 10-μM Y27632 in mTeSR1 medium for 1 h before dissociation into single cells using Accumax cell dissociation solution. Following dissociation, ~60,000–80,000 cells (seeding number needs be optimized for different hPSC lines) were seeded into Matrigel coated 12-well plate with 1-mL mTeSR1 medium with 10-μM Y27632 (old protocol, Figs. 4) or 1x CloneR (STEMCELL Technologies, Cat. No. 05888) (refined protocol, Fig. 5) (Day 0). Twenty-four hours later (Day 1), the medium was removed and 1 mL of pre-warmed ME stage medium was slowly added to the well. Forty-eight hours later (Day 3), ME stage medium was removed and 1 mL of UB-I stage medium was slowly added to the well. Twenty-fours hours later (Day 4), medium was changed to 1 mL of fresh UB-I medium again. After another 24 h (Day 5), UB-I medium was removed, and 1.5-2 mL of UB-II stage medium was slowly added to the well. Twenty-four hours later (Day 6), medium was changed to 1.5-2 mL of fresh UB-II medium. At Day 7, differentiated cells were dissociated into single cells following the standard Accumax dissociation method. For wild-type hESC line without any reporter, cell pellet after dissociation was resuspended in 250–400-μL cold FACS medium (1x PBS, 1X Pen Strep, 2% FBS) supplemented with a PE conjugated anti-human CD117(C-KIT) antibody (Biolegend, Cat. No.313204) at 1:200 dilution and incubated for 30 min on ice. The tube was gently tapped every 10 min to ensure mixing. After 30 min, 1-mL cold FACS medium was added into the tube. The tube was then centrifuged at $300 \times g$ for 5 min and the supernatant was carefully removed. The pelleted cells were resuspended with 300–500-μL cold FACS medium plus DAPI at 1:2000 ratio, placed through 40-μm cell strainer (Greiner bio-one, Cat. No. 542040) and transferred into a FACS tube on ice before FACS. C-KIT+ (PE) cells were then sorted out by FACS. For reporter cell lines (*PAX2*-mCherry/*WNT11*-GFP hESC line or *SOX9*-GFP hiPSC line), dissociated cells were either resuspended directly in 300–500-μL cold FACS medium plus DAPI at 1:2000 ratio, or first went through the C-KIT staining process described above before resuspended in FACS medium with DAPI.

These cells were then put through a 40-μm cell strainer and placed on ice before FACS. mCherry+ cells (from the *PAX2*-mCherry/*WNT11*-GFP hESC line), GFP+ cells (from the *SOX9*-GFP hiPSC line), or C-KIT+ (PE) cells (after C-KIT staining) were then sorted out by FACS. Upon FACS sorting of the mCherry+ or GFP+ or PE+ cells, these cells were collected in a 1.5-mL Eppendorf tube with 500-μL 10% FBS. The tube was centrifuged at $300 \times g$ for 5 min, and the supernatant was carefully removed. Cell pellet was then resuspended in an appropriate amount of hUBCM or hUBCM-v2 (with the addition of Y27632 at 10-μM final concentration), and cell density was measured by automatic cell counter. Approximately, 2000–20,000 cells were transferred into each well of a U-bottom 96-well low-attachment plate, and an appropriate amount of hUBCM or hUBCM-v2 (with 10-μM Y27632 for the first 24 h) was added to the well to make the final volume of 100 μL per well. After 24 h, UB cell aggregate was formed and embedded into an 8-μL cold Matrigel droplet in another well of the U-bottom 96-well low-attachment plate and cultured in a 37 °C incubator following the same embedding procedure described above (with hUBCM or hUBCM-v2). After ~7–10 days of culturing, epithelial tip structures could be seen budding out from the aggregate. *WNT11*-GFP expression was induced within 10 days in the *PAX2*-mCherry/*WNT11*-GFP hESC line-derived organoid. These tip structures were dissected out and re-embedded into Matrigel to continue culture. From then, the iUB organoid was established and can be passaged stably following the procedures below.

**Human UB organoid expansion and passaging**. During the culture, human UBCM was renewed with fresh medium every 2 days. Both human UB organoid from primary RET+ UPC and iUB organoid from hPSCs were passaged every 6–10 days depending on the size. The passaging methods (manual or as single cells) are the same as defined above in the mouse UB organoid section, with the change of using hUBCM or hUBCM-v2 instead of mUBCM.

**CD differentiation**

*mCD differentiation.* Mouse UB organoid was passaged at Day 5 of expansion as single cells, and 2000 cells were seeded for continuing expansion. At Day 10 of mUB expansion, mUBCM was removed and 150-μL 1x PBS was added and removed to wash the organoid. One hundred and fifty microliters of mouse CDDM was then added to initiate mCD differentiation (mCD differentiation Day 0). The organoid was cultured in a 37 °C incubator and medium was changed every 2 days or daily as needed if the medium turns yellow/orange for a total of 7 days. No passage of the organoid was needed. At mCD differentiation Day 7, the mCD organoid was harvested for analyses.

*Human CD differentiation.* After human UB organoid expansion was stabilized (at least 25 days post FACS when UB organoid were growing stably) and reached an appropriate size (at least 900-μm diameter), hUBCM was removed and 150-μL 1x PBS was added and removed to wash the organoid. One hundred and fifty microliters of hCDDM was added to start hCD differentiation (hCD differentiation Day 0). The organoid was cultured in 37 °C incubator and medium was changed every 2 days or daily if needed if the medium turns yellow/orange for a total of 14 days. At hCD differentiation Day 14, hCD organoid was harvested for analyses.

**Mouse engineered kidney generation (Refer to Supplementary Methods for more details)**. A small piece of mUB organoid was manually dissected out and inserted into a microdissected hole on a 3D cultured mNPC aggregate to generate a engineered kidney precursor. This precursor was then carefully transferred into a well of a U-bottom 96-well low-attachment plate with 100-μL kidney reconstruction medium (APEL2 + 0.1-μM TTNPB) with 10-μM Y27632 and cultured in 37 °C incubator (Day 0). After 24 h (Day 1), the engineered kidney precursor was transferred onto a six-well transwell insert membrane with kidney reconstruction medium for continuring culture.

**RNA sequencing**. Adult mCD cells were FACS isolated (*Hoxb7*-Venus+) from adult (~2 month old) *Hoxb7*-Venus mouse kidneys. All samples were collected and lysed in TRIzol reagent and stored under −80 °C. Total RNA was extracted using the Direct-zol RNA MicroPrep Kit (Zymo). cDNA library was prepared using the KAPA Stranded mRNA-Seq Kit (KAPA Biosystems). RNA sequencing was performed by the Children's Hospital Los Angeles Molecular Pathology Genomics Core.

**Gene editing in UB organoids (Refer to Supplementary Methods for more details)**. Lentiviral infection was used to edit genes in mUB cells. One hundred microliters mixture of 1–5x virus, 10-μM Polybrene, and UBCM was added to the U-bottom 96-well low-attachment plate well with UB single cells suspension. The UBs and virus were centrifuged together at 800 g for 15–30 min for spinfection[70] at room temperature. The infected UB cells were then washed, aggregated overnight, embedded in Matrigel, and cultured in UBCM in 37 °C incubator following standard UB organoid culture procedures described above. Appropriate antibiotic was added to the culture to select for UB cells that have been successfully infected.

**RNA-seq data analysis**. RNA sequencing data was analyzed using Partek Flow (version 10.0.21.0411), including published dataset of IPCs and NPCs[36], ureteric tip and trunk cells[35]. FASTQ files were trimmed from both ends based on a minimum read length of 25 bps and a shred quality score of 20 or higher. Reads were aligned to GENCODE mm10 (release M24) using STAR 2.5.3a. Aligned reads were quantified to the Partek E/M annotation model. Gene counts were normalized by adding 1 then by TMM values. All TMM values can be found in Supplementary Data 1. Samples were filtered to include differentially expressed genes of UB tip compared to UB trunk, with false discovery rate ≤0.01, fold change <−4 or >4, total counts ≥10, and $p$ value < 0.05, resulting in 1413 UB tip/trunk signature genes (Supplementary Data 2). Then, hierarchical clustering was produced by clustering samples and features with average linkage cluster distance and Euclidean point distance. PCA was performed using the EDASeq R/Bioconductor packages and the plot was rendered with the ggplot2 R package.

**Statistics and reproducibility**. All statistical analysis shown in figures are specified in the corresponding legends. All statistical analysis was performed using Prism 8 (version 8.2.1). All bright field and immunofluorescence micrographs shown are representative images selected from at least three independent experiments all showing similar results. Genetic engineering of hESCs and DNA gel electrophoresis (Supplementary Fig. 4a–e) were performed once.

**Reporting summary**. Further information on research design is available in the Nature Research Reporting Summary linked to this article.

## Data availability
RNA-seq data have been submitted to Gene Expression Omnibus (GEO) with accession number "GSE149109 [https://www.ncbi.nlm.nih.gov/geo/query/acc.cgi?acc=GSE149109]." For GUDMAP/RBK resources, visit https://www.gudmap.org. Key resources such as antibodies, chemicals, recombinant proteins, genotyping, and qRT-PCR primers are provided in Supplementary Tables 3–5. All other relevant data supporting the key findings of this study are available within the article and its Supplementary Information files or from the corresponding author upon reasonable request. A reporting summary for this Article is available as a Supplementary Information file. Source data are provided with this paper.

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

## Acknowledgements

We would like to thank Jeffrey Boyd and Bernadette Masinsin of the USC Flow Cytometry Facility for FACS; Seth Ruffins of the USC Optical Imaging Facility for help with microscopy; Dejerianne Ostrow and David Ruble of the Children's Hospital Los Angeles Molecular Pathology Genomics Core for RNA-seq; Meng Li, Yibu Chen, and Eddie Loh of the USC Norris Medical Library Bioinformatics Service for help with the RNA-seq computational analysis; Haruhiko Akiyama and Juan Carlos Izpisua Belmonte for sharing the *Sox9*-GFP mice, Naoki Nakayama for sharing the *SOX9*-GFP hiPSC line; Dr Melissa L. Wilson (Department of Preventive Medicine, University of Southern California) and Family Planning Associates for coordinating fetal tissue collection; and Cristy Lytal for helping with editing the manuscript. This work is supported by departmental startup funding and USC/UKRO Kidney Research Center funding to Z.L. J.C. was supported by CIRM Bridges Program. Z.Z. was supported by USC Stem Cell Challenge Award. Work in A.P.M.'s laboratory was supported by a Grant from the NIDDK (DK054364). E.A.R. was supported by an F31 fellowship (DK107216).

## Author contributions

Z.Z., B.H., A.P.M., and Z.L. designed the study. Z.Z., B.H., R.K.P., Y.L., J.C., T.P., E.A.R., A.D.K. and J.A.M. performed experiments. A.C.V. and M.E.T. performed RNA-seq computational analysis. M.E.T and B.H.G. provided human fetal kidney samples. N.M. P.-S., K.R.H., and A.P.M. provided reagents and helpful discussions. Z.Z., B.H., and Z.L. wrote the manuscript. Z.Z., B.H., J.Y., N.M.P.-S., K.R.H., A.P.M., and Z.L. edited the manuscript.

## Competing interests

Patent application has been filed on April 27, 2020 by University of Southern California on behalf of inventors Z.Z., B.H., A.P.M., and Z.L. for the mouse and human UB organoid generation, expansion, and CD organoid differentiation systems described in this study. The other authors declare no competing interests.
