## [Peer Review File · Nature Communications]

Reviewers' Comments:

Reviewer #1:

Remarks to the Author:

In the manuscript entitled: Generation of Patterned Kidney Organoid that Recapitulates Adult Kidney Collecting System from Expendable Ureteric Bud Progenitors, the authors provide detailed description of a novel protocol to develop a mammalian CD organoid.

The paper is significant as development of collecting duct system is essential to develop an organoid/organ with all features of the functional mammalian kidney that can serve as a model for diseases and/or for transplantation. The authors provide approaches to generate UB organoids from mouse and human fetal kidneys as well as human pluripotent stem cells, providing multiple options to pursue. Development of a reporter for UPC population in mice, allowed identification of a cocktail that allowed propagation of the UPCs and expansion more than 100 thousand-fold.

1. The transcriptome analysis is a strength. Do the authors see any drawbacks of prominent UB tip molecular signature and not stalk cells? How will this ultimately affect ability to generate a robust disease model and/or fully differentiated organoid?
2. Can the authors expand on likely identity of clonally derived self-organizing organoids (~30% of cells) that did not retain Wnt11-RFP expression? Are these cells more stalk-like or are they intermediate cell types?
3. The development of UB organoids from 3 different sources is very powerful. There are not as many data regarding the hiPSC and hESC -additional information and characterization would strengthen the paper as these source cells would be very useful. It is recognized that COVID-19 shutdown prevented some of this work.
4. The identification of principal cells and intercalated cells with at least some appropriate distribution and expression of transporters and molecule is exciting. How variable is the percent contribution of different cell types? Is there any specific distribution? Can the authors comment on why they believe these cell types differentiate in a non-filtering kidney?

Reviewer #2:

Remarks to the Author:

Most attention so far in the kidney regenerative field has been going to the nephrogenic lineage which can now be routinely studied in vitro in monocultures and organoids. In this manuscript describe their work to get the epithelial lineage / collecting duct system top the same level. They generate a Wnt11-myrTagRFP-IRES-CE (Wnt11-RFP) for labeling, following and tracing Wnt11-positive UPCs. The use this model to optimize culture conditions for T-shaped UBs to come to UBCM (UB culture medium) and show that these cultures work for different strains. They show that they can reform UB organoids although only 3-5% maintains the RFP signal. Next they screen for conditions to mature the UB organoids to CDs using a qRT-PCR approach based on Aqp2 and Foxi1 expression as markers for PC and IC cells. This lead to the CD differentiation medium (CDDM) composition. The authors show they can form synthetic (I prefer the term engineered) kidneys by combing in vitro cultured NPCs with in vitro cultured UB organoids and get nice branching organoids with nicely connected nephrons expression markers for several nephron lineages. Finally, it was shown that when UB organoids can be made single cell and at that stage genetically modified (overexpression or gene edited) using lentivirus. Next the authors move on to the human system. They identify human UBCM, different from mouse UBCM. They generate a PAX2-mCherry WNT11-GFP reporter human ES cell line and differentiate them to UB-like cells with published methods but found limited efficiency, but using this reporter line improve the method further. They show they can make iUBs using their optimized conditions which they could mature further with CDDM as before. The same iUBs they could make from iPSCs.

The manuscript describes solid data, but the following issues hamper my enthusiasm for it to support publication of the current manuscript in Nature Comm.

1. The exact origin and nature of the Wnt11-RFP mice is a complete mystery to me. They are

described as 'has been generated' without a reference suggesting they were made in this work. However, in the acknowledgment they're mentioning as a gift of Andy McMahon, who is also a co-author of the manuscript. If this model has not been published, a lot more (technical) detail should be given here. How was it made? What did the construct look like? What basic characterization of the model has been done? What does the model look like in normal mouse embryonic kidneys? This model is the basis of the whole manuscript, the reader should be able to judge if the bass is correct. I'm not suggesting here it might not be, but it should simply be available.

2. The development of the UBCM is presented in the Results section as something completely done here, although any details of compounds tested or making up the medium are here completely lacking. Only in the Methods it is becoming clear that it is in fact merely an optimization of the of the conditions previously published by Yuri et al (2017). The work of Tagushi and Nishinakamura (2017) in Cell Stem Cell who came up with convincing UB culture protocols from single cells is not even mentioned in this context in sup table 2.. There is nothing wrong with optimizing previously described conditions, there is a lot wrong with hiding this fact in the Methods and in the Results presenting it is new work. The origin and (final) composition of the UBCM should be made perfectly clear in the Results section.

3. It is completely unclear to me where fig. 1e is coming from and what it shows other than that it is quantification of the imaging data in fig 1d, and sup fig 2a. 2b? What is meant with 'positive staining percentage' (y-axis) and how does this relate to the markers mentioned for the different bars? It doesn't seem to be a ratio as some of the bars represent 3 markers. How were the images quantified? How were the calculations done? Nothing is described.

4. The authors compare the limited capability of maintaining RFP signal in the single-cell UB organoids to the comparable efficiency with Lgr5+ intestinal stem cells and intestinal organoids. However, in the latter case it was later shown that preserving the interaction between stem cell and Paneth cell greatly improved this efficiency, providing important extra biological insights about the intestinal stem cell niche. Could a comparable situation be in play here? This should at least be discussed.

5. The imaging data in fig 2c-e on the maturation of PC and IC cells looks interesting, but it cannot be properly interpreted and judged without a normal mouse kidney next to it stained for the same markers. Only this can show in how far the patterns seen in the CD organoids are normal or still have a (long) way to go and without it certainly normal patterning cannot be claimed.

6. I'm positively surprised by how well the synthetic kidneys look in the absence of the stromal lineage. Previously it was shown through cell ablation studies that the stromal lineage is absolutely essential for the nephrogenic lineage to develop, and the (again ignored) work of Tagushi and Nishinakamura (2017) who already demonstrated such synthetic kidneys indeed needed the addition of SP progenitors. The authors should at least discuss this remarkable difference.

7. The gene editing and overexpression data is in my opinion of limited value, others have shown comparable approaches using normal kidneys (though this work is not mentioned) so it is not very surprising this also works in this context. For me it doesn't add much to the overall manuscript.

8. The hUBCM conditions appear much more robust than mouse (for instance 70 days vs 20). Can the authors explain this? Has the hUBCM been tested on mouse to see if it also works better there?

Reviewer #3:

Remarks to the Author:

With interest I have read the manuscript by Zeng et al. The authors describe two major findings. First, the establishment of branching and expanding ureteric bud (UB) organoids from both mouse and human primary UB progenitors, as well as from human embryonic and induced pluripotent stem cells. Second, conditions that allow effective differentiation towards principal and intercalated cells. These findings are interesting and advance the field, although the first carries somewhat less

novelty given the articles by Taguchi and Nishinakamura (Cell Stem Cell 2017) and Yuri et al. (Stem Cell Reports 2017).

Below several specific questions and remarks.

- Human induced pluripotent stem cells seem the most favorable source of UB organoids in this article, yet these are only characterized by SOX9 expression (which is also expressed in nephron precursors during various stages of nephrogenesis). Why not include these in the qPCR and immunocytochemistry characterization?
- The authors claim efficient introduction of genetic knockouts using a mouse that constitutively expresses Cas9. First, using a single gRNA (instead of a mixture of 3 different gRNAs) would give a more accurate estimation of the knock-out efficiency and reduce unwanted off-target effects. Second and more important, it would be much more informative (and useful) if gene editing efficiency is shown upon transfection with Cas9 in mouse and human organoids, rather than in mice that constitutively express Cas9. Third, the power of these organoids and gene-editing would be significantly substantiated if an example of a clinically relevant gene knockout is given (for example a gene involved in congenital anomalies of the kidney and urinary tract).

Minor points:

- The authors describe that branching mouse organoids were cultured up to ~15-20 days. Indicate what happens after 20 days: did organoids stop growing? (fig 1c seems suggestive of this) Were there any changes in growth speed and organoid characteristics over time (for both mouse and human organoids)?
- What happens after Wnt11 expression is lost after 10 days of mouse UB organoid culture? Do organoids differentiate towards UB trunk or collecting duct cells? It would be interesting to add organoids at the latest possible timepoint to the RNA sequencing in fig 1f, as well as differentiated organoids and mature collecting duct.
- Fig 2.c-e: Was differentiation performed on intact large branching organoids or rather small fragments of these? (the schematic in fig 2a suggests the first, whereas fig 2c-e suggest the latter). Please elucidate and also provide brightfield pictures of whole differentiated organoids.
- Add side-by-side Z-stack overlay pictures with all channels and only the green channel (KRT8 + PODXL) to fig. 3c to more clearly show if there is extensive branching and a common collection point for all branches in the assembled kidney.
- Fig 4d + 4f: Indicate after how many days of culture/passages organoids were analyzed. More importantly, provide similar expression results for human organoids cultured for several timepoints over the course of 70 days to give insight in the development of the organoid cultures over time.
- The authors report that the NOTCH ligand JAG1 preferentially increased intercalated cell-specific FOXI1. Interestingly, activation of NOTCH signaling was reported to mediate a switch from intercalated cells towards principal cells in for example the article by Park et al. Science 2018. It would be interesting to hear how the authors reflect on these diverging findings.

Reviewer #1 (Remarks to the Author):

In the manuscript entitled: Generation of Patterned Kidney Organoid that Recapitulates Adult Kidney Collecting System from Expendable Ureteric Bud Progenitors, the authors provide detailed description of a novel protocol to develop a mammalian CD organoid.

The paper is significant as development of collecting duct system is essential to develop an organoid/organ with all features of the functional mammalian kidney that can serve as a model for diseases and/or for transplantation. The authors provide approaches to generate UB organoids from mouse and human fetal kidneys as well as human pluripotent stem cells, providing multiple options to pursue. Development of a reporter for UPC population in mice, allowed identification of a cocktail that allowed propagation of the UPCs and expansion more than 100 thousand-fold.

We thank the reviewer for the recognition of the significance of our work.

1. The transcriptome analysis is a strength. Do the authors see any drawbacks of prominent UB tip molecular signature and not stalk cells? How will this ultimately affect ability to generate a robust disease model and/or fully differentiated organoid?

We thank the reviewer for recognition of our transcriptome analysis. In our revised manuscript, we further strengthened this part by including UB organoids cultured for 20 days into our RNA-seq analysis, as well as our CD organoids and primary mouse CD freshly isolated from adult kidney of Hoxb7-Venus mice (Fig. 1f and 2i). As shown by PCA, all the UB organoids are clustered closely to the primary UB tip population (Fig. 1f). It also shows a clear separating of CD organoids from the immature UB tip and UB organoid populations, and similar grouping to UB trunk and primary mouse CD (Fig. 2i), supporting an expected progenitor identity of the UB organoids and maturation of the CD organoids. Consistent with Wnt11-RFP reporter expression, during the first 10 days of culture, most of the UB tip genes are stably maintained as revealed by the TMM values of the RNA-seq dataset (all TMM values can be found in Supplementary Table 10), and none of the stalk genes or mature CD marker genes are expressed throughout the culture period in our UB organoid. However, a gradual and significant reduction of Lhx1 expression (averaged TMM values: at day 5, 1902.3; at day 10, 285.5; and at day 20, 28.3) was observed, indicating some differences at the molecular level after extended culture. Importantly, these UB organoids can differentiate into well patterned CD organoids, demonstrating that fully differentiated organoid formation might not be significantly affected by certain changes at the molecular level (Fig. 2c-i, and Supplementary Fig. 3g-o). In our revised manuscript, we also provide evidence to show that a robust disease model can be generated from both mouse and human UB organoids (Fig. 6 and Supplementary Fig. 7 a-f). Briefly, we have used CRISPR/Cas9 to knock out Ret/RET in mouse and human UB organoids. RET plays key roles in mediating UB branching morphogenesis and mutations of RET lead to severe congenital anomalies of kidney and urinary tract (CAKUT) phenotypes in human patients. Upon highly efficient Ret/RET gene knockout (Fig. 6c, d, g, h), both mouse and human UB organoids showed strong phenotype of impaired branching morphogenesis (Fig. 6b, f), mimicking that of Ret knockout in vivo. At the molecular level, UB marker gene expression is significantly affected (Fig. 6e, i), and interestingly, we observed different outcomes to validated mouse targets in the human UB. These results further strengthen the utility of our UB organoid model in the study of kidney development and disease, and potentially reveal distinct insights into human development.

2. Can the authors expand on likely identity of clonally derived self-organizing organoids (~30% of cells) that did not retain Wnt11-RFP expression? Are these cells more stalk-like or are they intermediate cell types?

To get more insights into the potential identity of these Wnt11-RFP⁻ organoids, we have compared the gene expression of these organoids with the Wnt11-RFP⁺ organoids by qRT-PCR. As shown in the figure on the right, consistent with the loss of the Wnt11-RFP signal in the Wnt11-RFP⁻ organoid, Wnt11 was also decreased at the mRNA level. Interestingly, none of the mature markers, Aqp2 and Foxi1, was elevated. These results

suggest that these Wnt11-RFP organoids are more likely to be an intermediate cell type, but are not spontaneously differentiating into a more mature stalk-like fate. This is also consistent with our RNA-seq results in which after extended culture period of time, the UB organoids lost Wnt11 expression, but never spontaneously induced any stalk or CD markers.

3. The development of UB organoids from 3 different sources is very powerful. There are not as many data regarding the hiPSC and hESC -additional information and characterization would strengthen the paper as these source cells would be very useful. It is recognized that COVID-19 shutdown prevented some of this work.

We thank the reviewer for recognition of the interruption of our research by COVID-19, and for this insightful suggestion to strengthen our study. In our revised manuscript, we have carried out a comprehensive characterization of our iUB organoids derived from hESC (Fig. 5, and Supplementary Fig. 5) and hiPSC (Supplementary Fig. 6).

In this revised manuscript, we further refined our iUB differentiation protocol, resulting in a new protocol that we expect to be broadly useful for any valid hESC or hiPSC line. Briefly, with this new protocol, we can 1) generate pure UPCs within 12 days starting from hPSCs; 2) the UPCs can be expanded homogeneously in the form of branching iUB organoids; and 3) the new protocol does not rely on specific reporter hPSC lines to purify the precursor cells, thus can be applied to any given hPSC line to derive iUB organoid (Fig. 5a). As proof-of-concept, using this refined protocol, we have successfully derived long-term expandable branching iUB organoids from 2 hESC lines (Fig. 5, and Supplementary Fig. 5), and 1 hiPSC line (Supplementary Fig. 6). At the time of this revised manuscript submission, all the UB organoids have been stably expanded for more than 70 days (a 10^8 - 10^9 fold increase), maintaining stable expression of UB marker genes comparable or higher than that of human fetal kidney sample (Fig. 5b, k, and Supplementary Fig. 5c, h, 6c, d). Importantly, at a later culture time point (day 66), the UB marker gene expression did not show any decrease as compared to early culture time points (day 33 and day 49), showing the robustness of our UB culture system in supporting long-term expansion of these organoids with high quality (Fig. 5b). More importantly, from either whole-mount staining or section staining, we have shown consistent results indicating a homogeneous expression of broad UB lineage markers (KRT8, PAX2, PAX8, GATA3 and CDH1), and UB tip markers (RET and SOX9) in all of these iUB organoids, indicating the progenitor identity of our iUB organoids (Fig. 5c-f, and Supplementary Fig. 5d-f, 6e-h). This feature of our iUB organoids separates us from other related studies in which a mixture of UB tip and trunk cells are generated in the organoids (see also Supplementary Table 2 for a comparison). Consistently, various PC markers (AQP2, AQP3, AQP4) and IC marker (FOXI1) were elevated hundreds to thousands of folds as revealed by qRT-PCR (Fig. 5g) and immunostaining further confirmed the expression of AQP3 in the iCD organoids, suggesting the maturation of the iUB organoids into collecting duct cell types. We would like to point out that we have tried to carry out a comprehensive characterization of our iCD organoids by immunostaining. However, after rigorous testing in human fetal kidney section samples, only AQP3 and FOXI1 antibodies were confirmed to be compatible for immunostaining (Supplementary Fig. 7g-m, note that several antibodies used from related studies somehow did not give a specific signal in human fetal kidney section). We were able to detect AQP3 expression in 20-30% of the cells in the iCD organoid. However, even though FOXI1 expression is dramatically induced as shown by qRT-PCR, we could not see clear signal of FOXI1 by immunostaining. This suggested that different from our well-established mouse CD organoid differentiation protocol, future efforts might be needed to further refine our iCD differentiation protocol for further maturation of iUB organoids into IC cell types.

4. The identification of principal cells and intercalated cells with at least some appropriate distribution and expression of transporters and molecule is exciting. How variable is the percent contribution of different cell types? Is there any specific distribution? Can the authors comment on why they believe these cell types differentiate in a non-filtering kidney?

We thank the reviewer for the positive comments on our innovation of generating spatially-patterned PC and IC cell types in our mouse CD organoid. We have carried out dozens of CD organoid differentiation experiments in our lab so far. Starting from a very stable UB organoid culture, we found that the differentiation into CD organoid is also very stable, with limited variations between experiments (Fig. 2c-h, and Supplementary Fig. 3g-o). As quantified in Fig. 2h, AQP2⁺ PC are usually 40-45%, while the FOXI1⁺ IC are usually 50-55%. It is

interesting that in our CD organoids, the IC portion appears to be more than PC, which is different from that in the postnatal mouse kidney. It is likely due of the use of DAPT, a small molecule inhibitor of Notch signaling, in our CD differentiation medium. Inhibition of Notch is well known to favor IC over PC fate in vivo (Jeong et al., 2009; Mukherjee et al., 2019; Park et al., 2018; Werth et al., 2017). It is an interesting question of why PC and IC form in vitro without filtration. During kidney development, the forming collecting duct starts to express PC markers as early as E13.5-E15.5, with IC markers expressed abundantly only at later embryonic stage (Werth et al., 2017). This observation suggests that the initial specification of PC/IC cell types might not rely on filtration. However, we speculate that filtration might be needed for further maturation of these cells. In supporting of this, according to our own RNA-seq data obtained from E16.5 mouse kidney (Rutledge et al., 2017), several PC and IC marker genes are already expressed in the UB trunk at comparable levels to that of an adult kidney (see TMM values in Supplementary Table 10). Interestingly, at this stage, key transporters in the IC, Slc4a1 (Ae1) and Slc26a4 (Pendrin), are still expressed at low levels compared to adult kidney. Our CD organoid appears to be in-between the UB trunk and the adult kidney, which has low Slc4a1, like the UB trunk, but showed dramatically elevated Slc26a4, comparable to that of an adult kidney. It will be interesting to test whether filtration promotes further maturation towards adult kidney and our in vitro accessible CD organoid might provide a unique platform for that purpose.

--

Reviewer #2 (Remarks to the Author):

Most attention so far in the kidney regenerative field has been going to the nephrogenic lineage which can now be routinely studied in vitro in monocultures and organoids. In this manuscript describe their work to get the epithelial lineage / collecting duct system top the same level. They generate a Wnt11-myrTagRFP-IRES-CE (Wnt11-RFP) for labeling, following and tracing Wnt11-positive UPCs. The use this model to optimize culture conditions for T-shaped UBs to come to UBCM (UB culture medium) and show that these cultures work for different strains. They show that they can reform UB organoids although only 3-5% maintains the RFP signal. Next they screen for conditions to mature the UB organoids to CDs using a qRT-PCR approach based on Aqp2 and Foxi1 expression as markers for PC and IC cells. This lead to the CD differentiation medium (CDDM) composition. The authors show they can form synthetic (I prefer the term engineered) kidneys by combing in vitro cultured NPCs with in vitro cultured UB organoids and get nice branching organoids with nicely connected nephrons expression markers for several nephron lineages. Finally, it was shown that when UB organoids can be made single cell and at that stage genetically modified (overexpression or gene edited) using lentivirus.

Next the authors move on to the human system. They identify human UBCM, different from mouse UBCM. They generate a PAX2-mCherry WNT11-GFP reporter human ES cell line and differentiate them to UB-like cells with published methods but found limited efficiency, but using this reporter line improve the method further. They show they can make iUBs using their optimized conditions which they could mature further with CDDM as before. The same iUBs they could make from iPSCs.

The manuscript describes solid data, but the following issues hamper my enthusiasm for it to support publication of the current manuscript in Nature Comm.

We thank the reviewer for recognition of the significance of our work and to recognize our work as “solid”. Based on the reviewer’s suggestion, we have changed our term from “synthetic kidney” to “engineered kidney” throughout the manuscript.

1. The exact origin and nature of the Wnt11-RFP mice is a complete mystery to me. They are described as ‘has been generated’ without a reference suggesting they were made in this work. However, in the acknowledgment they’re mentioning as a gift of Andy McMahon, who is also a co-author of the manuscript. If this model has not been published, a lot more (technical) detail should be given here. How was it made? What did the construct look like? What basic characterization of the model has been done? What does the model look like in normal mouse embryonic kidneys? This model is the basis of the whole manuscript, the reader should be able to judge if the bass is correct. I’m not suggesting here it might not be, but it should simply be available.

We apologize for not including the reference to this mouse strain. It has been published by McMahon's group in 2017 (Rutledge et al., 2017). This is a knockin mouse strain in which a myrTagRFP is under the control of endogenous Wnt11 promoter. As shown in Rutledge et al., 2017, Wnt11-RFP reporter is expressed exclusively in the tip region of the branching UB during kidney development. This useful reporter line has been utilized to profile UB tip and UB trunk gene expression by RNA-seq, which leads to the identification of tip and trunk signature genes, validating the robustness of this mouse model. We have added this reference in the Results section when describing the use of this mouse strain, and provided in the Methods section the ordering information for this mouse strain from the Jackson Laboratory (JAX, #01683).

2. The development of the UBCM is presented in the Results section as something completely done here, although any details of compounds tested or making up the medium are here completely lacking. Only in the Methods it is becoming clear that it is in fact merely an optimization of the of the conditions previously published by Yuri et al (2017). The work of Taguchi and Nishinakamura (2017) in Cell Stem Cell who came up with convincing UB culture protocols from single cells is not even mentioned in this context in sup table 2.. There is nothing wrong with optimizing previously described conditions, there is a lot wrong with hiding this fact in the Methods and in the Results presenting it is new work. The origin and (final) composition of the UBCM should be made perfectly clear in the Results section.

We apologize for the confusion to the reviewer. We did not intentionally try to make the impression that the culture was identified without referencing to previous studies. We had put all the data in Supplementary Fig. 1, and all the details in the Methods part so that the reader would be able to see the most exciting discovery in the main Fig. 1 for what our new culture system can offer, without being overwhelmed by a lot of trivial technical details at the beginning. We did not notice that it would make such a wrong impression. To avoid that, we have now clearly stated in the Results part that the work was built upon previous efforts towards the culture of UB. In addition to acknowledge the work from Yuri et al., we also cited the pioneering work from the Nigam's group from almost 2 decades ago. The origin and final composition of the UBCM is also made clear in the Results section as the reviewer suggested.

The mouse UB part from Taguchi and Nishinakamura's work in 2017 was focused on developing a directed differentiation protocol from mouse embryonic stem cells into the UB lineage. It is different from what we do in this study starting from primary E11.5 UB aimed at capturing a pure population of UB progenitor cells in culture. So we did not include their work in comparison in Supplemental Table 1 in our initial submission. We have added it to the comparison table per reviewer's request.

3. It is completely unclear to me where fig. 1e is coming from and what it shows other than that it is quantification of the imaging data in fig 1d, and sup fig 2a. 2b? What is meant with 'positive staining percentage' (y-axis) and how does this relate to the markers mentioned for the different bars? It doesn't seem to be a ratio as some of the bars represent 3 markers. How were the images quantified? How were the calculations done? Nothing is described.

The reviewer is correct. This graph in Fig. 1e is a quantification of the images from Fig. 1d and Supplementary Fig. 2a, b. We apologize for the confusion in presenting this data. We have modified our Methods part to include more details of how the quantification was performed:

"Whole-mount immunostaining images for mouse UB organoids, mouse CD organoids, or human UB organoids were used for the quantification of various marker gene expression. ImageJ software was used to count positive cells. 3 different fields of view per organoid were randomly selected to count the number of positively stained cell numbers (positive for marker genes) and total cell numbers (DAPI+). Percentage was calculated by the number of cells that are positive for different UB/CD marker genes divided by the total DAPI+ cell numbers. At least 500 cells in total were counted. Error bars represent standard derivation between different field views."

4. The authors compare the limited capability of maintaining RFP signal in the single-cell UB organoids to the comparable efficiency with Lgr5+ intestinal stem cells and intestinal organoids. However, in the latter case it was later shown that preserving the interaction between stem cell and Paneth cell greatly improved this efficiency, providing important extra biological insights about the intestinal stem cell niche. Could a

comparable situation be in play here? This should at least be discussed.

*We thank the reviewer for insightful comments. We agree that a comparable situation might play a role here as suggested by the reviewer. Like *Lgr5*⁺ cells are surrounded by Paneth cells, UB tip cells are also tightly attached next to each other in the developing kidney. This cell-cell contact found in vivo or in intact UB organoid culture is disrupted in the in vitro clonal expansion culture where tip cells are dispersed in our experimental setting. It is likely that cell-cell contact is important for maintaining the best tip identity, as aggregated UPCs, or manually passaged UB organoids as small cell clusters, can maintain *Wnt11*-RFP homogeneously. Better understanding of the cell-cell contact or potential additional paracrine signals might help further improve the culture, thus allowing the development of a more robust clonal expansion method, similar to what was found in the intestinal system. We have discussed this in the Discussion section.*

5. The imaging data in fig 2c-e on the maturation of PC and IC cells looks interesting, but it cannot be properly interpreted and judged without a normal mouse kidney next to it stained for the same markers. Only this can show in how far the patterns seen in the CD organoids are normal or still have a (long) way to go and without it certainly normal patterning cannot be claimed.

*Following reviewer's suggestion, we have provided immunostaining results for these markers from neonatal and adult mice (Supplementary Fig. 3k-n). Similar to our CD organoids, in mouse kidneys, *FOXI*⁺/*ATP6V1B1*⁺ ICs are interspersed in the *AQP2*⁺/*AQP3*⁺ PCs. We have carried out dozens of CD organoid differentiation experiments in our lab so far. Starting from a very stable UB organoid culture, we found that the differentiation into CD organoid is also very stable, with limited variations between experiments (Fig. 2c-h, and Supplementary Fig. 3g-o). However, we do see some differences between our CD organoid and a normal mouse kidney. As quantified in Fig. 2h, *AQP2*⁺ PC are usually 40-45%, while the *FOXI*⁺ IC are usually 50-55%. In our CD organoids, the IC portion appears to be more than PC, which is different from that in the postnatal mouse kidney. It is likely due of the use of DAPT, a small molecule inhibitor of Notch signaling, in our CD differentiation medium. Inhibition of Notch is well known to favor IC over PC fate in vivo (Jeong et al., 2009; Mukherjee et al., 2019; Park et al., 2018; Werth et al., 2017).*

6. I'm positively surprised by how well the synthetic kidneys look in the absence of the stromal lineage. Previously it was shown through cell ablation studies that the stromal lineage is absolutely essential for the nephrogenic lineage to develop, and the (again ignored) work of Tagushi and Nishinakamura (2017) who already demonstrated such synthetic kidneys indeed needed the addition of SP progenitors. The authors should at least discuss this remarkable difference.

We thank the reviewer for this insightful comment. Stromal population is essential for normal kidney development and UB branching, at least partly through the production of retinoic acid (RA) (Batourina et al., 2001; Mendelsohn et al., 1999; Rosselot et al., 2010). Considering this, in our engineered kidney culture, we have included small molecule TTNPB, an analog of RA, into our medium. Activation of RA signaling by TTNPB potentially replaced some important functions of the stromal population in our system. On the contrary, in Dr. Nishinakamura's study, they did not include any factor in their culture medium to activate RA signaling. Under this context, in the absence of stromal population, a dramatic defect in kidney development might be observed. We have discussed this in the Discussion section of our revised manuscript.

7. The gene editing and overexpression data is in my opinion of limited value, others have shown comparable approaches using normal kidneys (though this work is not mentioned) so it is not very surprising this also works in this context. For me it doesn't add much to the overall manuscript.

*We thank the reviewer for this insightful comment. We agree a technical proof-of-concept by overexpressing and knockout of GFP is of limited value because GFP is not related to kidney development or disease. To add more values to this manuscript in regard to genome editing and disease modeling, in our revised manuscript, we provide evidence to show that a robust disease model can be generated from both mouse and human UB organoids (Fig. 6 and Supplementary Fig. 7 a-f). Briefly, we have used CRISPR/Cas9 to knock out *Ret/RET* in mouse and human UB organoids. *RET* plays key roles in mediating UB branching morphogenesis and mutations of *RET* lead to severe congenital anomalies of kidney and urinary tract (CAKUT) phenotypes in human patients (Arora et al., 2021; Costantini, 2012; McMahon, 2016; Nicolaou et al., 2015; Skinner et al.,*

2008). Upon highly efficient *Ret/RET* gene knockout (Fig. 6c, d, g, h), both mouse and human UB organoids showed strong phenotype of impaired branching morphogenesis (Fig. 6b, f), mimicking that of *Ret* knockout in vivo. At the molecular level, UB marker gene expression is significantly affected (Fig. 6e, i), and interestingly, we observed different outcomes to validated mouse targets in the human UB. These results further strengthen the utility of our UB organoid model in the study of kidney development and disease, and potentially reveal distinct insights into human development.

8. The hUBCM conditions appear much more robust than mouse (for instance 70 days vs 20). Can the authors explain this? Has the hUBCM been tested on mouse to see if it also works better there?

*We agree with the reviewer that the hUBCM is more robust than the mouse one in supporting long-term expansion of the human UB organoid. Inspired by the reviewer's comment, we tried to refine both mouse and human culture conditions. First, we tried to use hUBCM on the mouse UB culture as suggested by the reviewer. However, we could not find any significant improvement. We reasoned that some intrinsic mechanisms that limit the ~10 days lifespan of mouse UPCs in vivo is still not overcome in our in vitro culture to extend its lifespan beyond 10 days. Interesting, when we removed Y27632 from the hUBCM (the mouse UBCM does not have it), the new hUBCM, which we termed hUBCM-v2, could induce fully programmed WNT11+ iUB from hPSCs with dramatically improved efficiency (compare hUBCM from Supplementary Fig. 4i and hUBCM-v2 from Supplementary Fig. 5b). With hUBCM-v2, in this revised manuscript, we further refined our iUB differentiation protocol, resulting in a new protocol that we expect to be broadly useful for any valid hESC or hiPSC line. Briefly, with this new protocol, we can 1) generate pure UPCs within 12 days starting from hPSCs; 2) the UPCs can be expanded homogeneously in the form of branching iUB organoids; and 3) the new protocol does not rely on specific reporter hPSC lines to purify the precursor cells, thus can be applied to any given hPSC line to derive iUB organoid (Fig. 5a). As proof-of-concept, using this refined protocol, we have successfully derived long-term expandable branching iUB organoids from 2 hESC lines (Fig. 5, and Supplementary Fig. 5), and 1 hiPSC line (Supplementary Fig. 6). At the time of this revised manuscript submission, all the UB organoids have been stably expanded for more than 70 days (a 10^8 - 10^9 fold increase), maintaining stable expression of UB marker genes comparable or higher than that of human fetal kidney sample (Fig. 5b, k, and Supplementary Fig. 5c, h, 6c, d). Importantly, at a later culture time point (day 66), the UB marker gene expression did not show any decrease as compared to early culture time points (day 33 and day 49), showing the robustness of our UB culture system in supporting long-term expansion of these organoids with high quality (Fig. 5b). More importantly, from either whole-mount staining or section staining, we have shown consistent results indicating a homogeneous expression of broad UB lineage markers (*KRT8*, *PAX2*, *PAX8*, *GATA3* and *CDH1*), and UB tip markers (*RET* and *SOX9*) in all of these iUB organoids, indicating the progenitor identity of our iUB organoids (Fig. 5c-f, and Supplementary Fig. 5d-f, 6e-h). This feature of our iUB organoids separates us from other related studies in which a mixture of UB tip and trunk cells are generated in the organoids (see also Supplementary Table 2 for a comparison). Consistently, various PC markers (*AQP2*, *AQP3*, *AQP4*) and IC marker (*FOXI1*) were elevated hundreds to thousands of folds as revealed by qRT-PCR (Fig. 5g) and immunostaining further confirmed the expression of *AQP3* in the iCD organoids, suggesting the maturation of the iUB organoids into collecting duct cell types.*

--

Reviewer #3 (Remarks to the Author):

With interest I have read the manuscript by Zeng et al. The authors describe two major findings. First, the establishment of branching and expanding ureteric bud (UB) organoids from both mouse and human primary UB progenitors, as well as from human embryonic and induced pluripotent stem cells. Second, conditions that allow effective differentiation towards principal and intercalated cells. These findings are interesting and advance the field, although the first carries somewhat less novelty given the articles by Taguchi and Nishinakamura (Cell Stem Cell 2017) and Yuri et al. (Stem Cell Reports 2017).

We thank the reviewer for recognition of the significance of our work.

Below several specific questions and remarks.

- Human induced pluripotent stem cells seem the most favorable source of UB organoids in this article, yet

these are only characterized by SOX9 expression (which is also expressed in nephron precursors during various stages of nephrogenesis). Why not include these in the qPCR and immunocytochemistry characterization?

We thank the reviewer for this important suggestion. In addition to the iPSC-derived iUB organoid, in our revised manuscript, we have carried out a comprehensive characterization of our iUB organoids derived from both hESC (Fig. 5, and Supplementary Fig. 5) and hiPSC (Supplementary Fig. 6).

In this revised manuscript, we further refined our iUB differentiation protocol, resulting in a new protocol that we expect to be broadly useful for any valid hESC or hiPSC line. Briefly, with this new protocol, we can 1) generate pure UPCs within 12 days starting from hPSCs; 2) the UPCs can be expanded homogeneously in the form of branching iUB organoids; and 3) the new protocol does not rely on specific reporter hPSC lines to purify the precursor cells, thus can be applied to any given hPSC line to derive iUB organoid (Fig. 5a). As proof-of-concept, using this refined protocol, we have successfully derived long-term expandable branching iUB organoids from 2 hESC lines (Fig. 5, and Supplementary Fig. 5), and 1 hiPSC line (Supplementary Fig. 6). At the time of this revised manuscript submission, all the UB organoids have been stably expanded for more than 70 days (a 10^8 - 10^9 fold increase), maintaining stable expression of UB marker genes comparable or higher than that of human fetal kidney sample (Fig. 5b, k, and Supplementary Fig. 5c, h, 6c, d). Importantly, at a later culture time point (day 66), the UB marker gene expression did not show any decrease as compared to early culture time points (day 33 and day 49), showing the robustness of our UB culture system in supporting long-term expansion of these organoids with high quality (Fig. 5b). More importantly, from either whole-mount staining or section staining, we have shown consistent results indicating a homogeneous expression of broad UB lineage markers (KRT8, PAX2, PAX8, GATA3 and CDH1), and UB tip markers (RET and SOX9) in all of these iUB organoids, indicating the progenitor identity of our iUB organoids (Fig. 5c-f, and Supplementary Fig. 5d-f, 6e-h). This feature of our iUB organoids separates us from other related studies in which a mixture of UB tip and trunk cells are generated in the organoids (see also Supplementary Table 2 for a comparison). Consistently, various PC markers (AQP2, AQP3, AQP4) and IC marker (FOXI1) were elevated hundreds to thousands of folds as revealed by qRT-PCR (Fig. 5g) and immunostaining further confirmed the expression of AQP3 in the iCD organoids, suggesting the maturation of the iUB organoids into collecting duct cell types. We would like to point out that we have tried to carry out a comprehensive characterization of our iCD organoids by immunostaining. However, after rigorous testing in human fetal kidney section samples, only AQP3 and FOXI1 antibodies were confirmed to be compatible for immunostaining (Supplementary Fig. 7g-m, note that several antibodies used from related studies somehow did not give a specific signal in human fetal kidney section). We were able to detect AQP3 expression in 20-30% of the cells in the iCD organoid. However, even though FOXI1 expression is dramatically induced as shown by qRT-PCR, we could not see clear signal of FOXI1 by immunostaining. This suggested that different from our well-established mouse CD organoid differentiation protocol, future efforts might be needed to further refine our iCD differentiation protocol for further maturation of iUB organoids into IC cell types.

- The authors claim efficient introduction of genetic knockouts using a mouse that constitutively expresses Cas9. First, using a single gRNA (instead of a mixture of 3 different gRNAs) would give a more accurate estimation of the knock-out efficiency and reduce unwanted off-target effects. Second and more important, it would be much more informative (and useful) if gene editing efficiency is shown upon transfection with Cas9 in mouse and human organoids, rather than in mice that constitutively express Cas9. Third, the power of these organoids and gene-editing would be significantly substantiated if an example of a clinically relevant gene knockout is given (for example a gene involved in congenital anomalies of the kidney and urinary tract).

We thank the reviewer for this great advice which we agree can significantly strengthen our study. Following the reviewer's suggestions, in this revised manuscript, applying an efficient gene editing strategy to remove RET activity, we demonstrate genetically modified UB organoids can model congenital anomalies of kidney and urinary tract (CAKUT).

GDNF is a critical signal in both mouse and human UB culture. In vivo, GDNF secreted by metanephric mesenchyme cells surrounding UPC-containing branch tips signals via RET, with its co-receptor GFRA1, to maintain the UPC state and stimulate UB branching morphogenesis. Loss of the activity of these genes results in a CAKUT syndrome (Arora et al., 2021; Costantini, 2012; McMahon, 2016; Nicolaou et al., 2015; Skinner et

al., 2008). We employed CRISPR/Cas9 system to knock out Ret/RET in mouse and human UB organoids predicting as in vivo, UB organoid development in vitro would be Ret/RET-dependent (Fig. 6a). UB organoids were infected with lentivirus expressing Cas9 and one sgRNA targeting Ret/RET (in lentiCRISPR-v2 vector), while a control group received sgRNA without Cas9 (in lentiGuide-puro vector). Two independent sgRNAs were designed for Ret/RET gene knockout. As expected, the two control mouse UB organoids grew normally with maintained branching morphogenesis upon lentiviral infection and puromycin selection, while both Ret knockout (KO) UB organoids stopped branching (Fig. 6b). Whole-mount immunostaining of control and Ret KO UB organoids confirmed a dramatic reduction (more than 95%) of RET expression 6 days after lentiviral infection in the Ret KO UB organoids, demonstrating the successful removal of Ret (Fig. 6c, d). Consistent with the defect in branching morphogenesis in the Ret KO organoids, genes enriched in UB tip, Wnt11 and Lhx1, were reduced dramatically, and the expression of common UB lineage markers Pax2 and Gata3 were also decreased (Fig. 6e). Knockout of RET in the human iUB organoid also resulted in the arrest of branching morphogenesis in both of the RET KO organoids receiving two different sgRNAs (Fig. 6f) and a loss of RET immunostaining in more than 95% of cells with both sgRNAs (Fig. 6g, h, and Supplementary Fig. 7a-f). Interestingly, even though WNT11 and GFRA1 expression was significantly reduced in both RET KO iUB organoids, the expression of LHX, GATA3, PAX2, as well as ETV5 and SOX9, did not show consistent changes in both RET KO organoids (Fig. 6i). These results suggest that species-specific regulatory network downstream of Ret/RET might govern UB progenitor fate, consistent with previous observations of convergent and divergent mechanisms of nephrogenesis between mouse and human (Lindström et al., 2018a; Lindström et al., 2018b). Taken together, we provide a proof-of-concept for recapitulating kidney development and disease using mouse and human UB organoid models.

Minor points:

- The authors describe that branching mouse organoids were cultured up to ~15-20 days. Indicate what happens after 20 days: did organoids stop growing? (fig 1c seems suggestive of this) Were there any changes in growth speed and organoid characteristics over time (for both mouse and human organoids)?

Yes, the reviewer is correct. After 20 days, the mouse UB organoid stopped growing. In the first 10 days, the mouse UB organoids grew stably and very fast. From day 10 to day 20, the growth rate decreased gradually, until around day 20 when the UB organoid barely grew any more (Fig. 1c). On the contrary, under our most optimized hUBCM-v2 culture condition, the human UB organoids from various sources, grew very stably for at least 70 days at a constant growth rate (Supplementary Fig. 5c, h and 6c).

- What happens after Wnt11 expression is lost after 10 days of mouse UB organoid culture? Do organoids differentiate towards UB trunk or collecting duct cells? It would be interesting to add organoids at the latest possible timepoint to the RNA sequencing in fig 1f, as well as differentiated organoids and mature collecting duct.

Following reviewer's suggestions, we have included UB organoids cultured for 20 days into our RNA-seq analysis, as well as our CD organoids and primary mouse CD freshly isolated from adult kidney of Hoxb7-Venus mice (Fig. 1f and 2i). As shown by PCA, all the UB organoids are clustered closely to the primary UB tip population (Fig. 1f). It also shows a clear separating of CD organoids from the immature UB tip and UB organoid populations, and similar grouping to UB trunk and primary mouse CD (Fig. 2i), supporting an expected progenitor identity of the UB organoids and maturation of the CD organoids. Consistent with Wnt11-RFP reporter expression, during the first 10 days of culture, most of the UB tip genes are stably maintained as revealed by the TMM values of the RNA-seq dataset (all TMM values can be found in Supplementary Table 10). Interestingly, none of the stalk genes or mature CD marker genes are expressed throughout the culture period in our UB organoid even after after 20 days of culture when Wnt11-RFP expression is lost. These results suggest that the UB organoids do not spontaneously adopt a UB trunk or collecting duct fate after extended culture. It is likely that some of the components in our UBCM prevents the specification from the progenitor stage to the mature CD. It is also likely that other signals (as the ones identified in CDDM) are required to trigger the transition from UB to CD.

- Fig 2.c-e: Was differentiation performed on intact large branching organoids or rather small fragments of these? (the schematic in fig 2a suggests the first, whereas fig 2c-e suggest the latter). Please elucidate and also provide brightfield pictures of whole differentiated organoids.

The CD organoid differentiation is performed with whole large UB organoids. Following reviewer's suggestions, we have included bright field images showing the intact UB organoid and the differentiated CD organoid (Fig. 2c). Morphologically, a clear elongation is observed in the CD organoid, similar to what is observed in vivo. Furthermore, we have included a tile + Z-stack whole-mount immunostaining images of a whole intact CD organoid, from which an overview of the formation of large numbers of PC and IC is presented, demonstrating the robustness of our method (Fig. 2d).

- Add side-by-side Z-stack overlay pictures with all channels and only the green channel (KRT8 + PODXL) to fig. 3c to more clearly show if there is extensive branching and a common collection point for all branches in the assembled kidney.

To better present the dynamic branching morphogenesis process in the assembled kidney, we have included a time-course bright field and Hoxb7-Venus images of an assembled kidney generated from wild-type NPC and Hoxb7-Venus UB organoid (Fig. 3b). Clearly, extensive branching and a common collection point are observed as expected.

- Fig 4d + 4f: Indicate after how many days of culture/passages organoids were analyzed. More importantly, provide similar expression results for human organoids cultured for several timepoints over the course of 70 days to give insight in the development of the organoid cultures over time.

We apologize for not labeling the culture days in the original manuscript submitted. We have put the information in the figure legends in the revised version. These UB organoids were cultured for more than 50 days before the qRT-PCR was performed. Following the reviewer's suggestion, we have performed a time-course qRT-PCR for iUB organoid cultured for 33 days, 49 days and 66 days. As shown in Fig. 5b, at a later culture time point (day 66), the UB marker gene expression did not show any decrease as compared to early culture time points (day 33 and day 49), showing the robustness of our UB culture system in supporting long-term expansion of these organoids with high quality.

- The authors report that the NOTCH ligand JAG1 preferentially increased intercalated cell-specific FOXI1. Interestingly, activation of NOTCH signaling was reported to mediate a switch from intercalated cells towards principal cells in for example the article by Park et al. Science 2018. It would be interesting to hear how the authors reflect on these diverging findings.

We thank the reviewer for this comment. We also noticed this unexpected result. The JAG-1 we used is a small peptide fragment of JAG-1, but not the full-sized protein (<https://www.anaspec.com/products/product.asp?id=38606>). It was originally used in a paper describing the use of this peptide to culture intestinal stem cells (Wang et al., 2015). How this peptide works has not been fully characterized. We reasoned it is likely that this peptide may serve as a dominant negative form and antagonize Notch signaling, rather than activate it.

REFERENCES:

- Arora, V., Khan, S., El-Hattab, A.W., Dua Puri, R., Rocha, M.E., Merdzanic, R., Paknia, O., Beetz, C., Rolfs, A., Bertoli-Avella, A.M., *et al.* (2021). Biallelic Pathogenic GFRA1 Variants Cause Autosomal Recessive Bilateral Renal Agenesis. *J Am Soc Nephrol* *32*, 223-228.
- Batourina, E., Gim, S., Bello, N., Shy, M., Clagett-Dame, M., Srinivas, S., Costantini, F., and Mendelsohn, C. (2001). Vitamin A controls epithelial/mesenchymal interactions through Ret expression. *Nat Genet* *27*, 74-78.
- Costantini, F. (2012). Genetic controls and cellular behaviors in branching morphogenesis of the renal collecting system. *Wiley Interdiscip Rev Dev Biol* *1*, 693-713.
- Jeong, H.W., Jeon, U.S., Koo, B.K., Kim, W.Y., Im, S.K., Shin, J., Cho, Y., Kim, J., and Kong, Y.Y. (2009). Inactivation of Notch signaling in the renal collecting duct causes nephrogenic diabetes insipidus in mice. *J Clin Invest* *119*, 3290-3300.
- Lindström, N.O., Guo, J., Kim, A.D., Tran, T., Guo, Q., De Sena Brandine, G., Ransick, A., Parvez, R.K., Thornton, M.E., Baskin, L., *et al.* (2018a). Conserved and Divergent Features of Mesenchymal Progenitor Cell Types within the Cortical Nephrogenic Niche of the Human and Mouse Kidney. *J Am Soc Nephrol* *29*, 806-824.
- Lindström, N.O., McMahon, J.A., Guo, J., Tran, T., Guo, Q., Rutledge, E., Parvez, R.K., Saribekyan, G., Schuler, R.E., Liao, C., *et al.* (2018b). Conserved and Divergent Features of Human and Mouse Kidney Organogenesis. *J Am Soc Nephrol* *29*, 785-805.
- McMahon, A.P. (2016). Development of the Mammalian Kidney. *Curr Top Dev Biol* *117*, 31-64.
- Mendelsohn, C., Batourina, E., Fung, S., Gilbert, T., and Dodd, J. (1999). Stromal cells mediate retinoid-dependent functions essential for renal development. *Development* *126*, 1139-1148.
- Mukherjee, M., deRiso, J., Otterpohl, K., Ratnayake, I., Kota, D., Ahrenkiel, P., Chandrasekar, I., and Surendran, K. (2019). Endogenous Notch Signaling in Adult Kidneys Maintains Segment-Specific Epithelial Cell Types of the Distal Tubules and Collecting Ducts to Ensure Water Homeostasis. *J Am Soc Nephrol* *30*, 110-126.
- Nicolaou, N., Renkema, K.Y., Bongers, E.M., Giles, R.H., and Knoers, N.V. (2015). Genetic, environmental, and epigenetic factors involved in CAKUT. *Nat Rev Nephrol* *11*, 720-731.
- Park, J., Shrestha, R., Qiu, C., Kondo, A., Huang, S., Werth, M., Li, M., Barasch, J., and Suszták, K. (2018). Single-cell transcriptomics of the mouse kidney reveals potential cellular targets of kidney disease. *Science* *360*, 758-763.
- Rosselot, C., Spraggon, L., Chia, I., Batourina, E., Riccio, P., Lu, B., Niederreither, K., Dolle, P., Duyster, G., Chambon, P., *et al.* (2010). Non-cell-autonomous retinoid signaling is crucial for renal development. *Development* *137*, 283-292.
- Rutledge, E.A., Benazet, J.D., and McMahon, A.P. (2017). Cellular heterogeneity in the ureteric progenitor niche and distinct profiles of branching morphogenesis in organ development. *Development* *144*, 3177-3188.
- Skinner, M.A., Safford, S.D., Reeves, J.G., Jackson, M.E., and Freemerman, A.J. (2008). Renal aplasia in humans is associated with RET mutations. *Am J Hum Genet* *82*, 344-351.
- Wang, X., Yamamoto, Y., Wilson, L.H., Zhang, T., Howitt, B.E., Farrow, M.A., Kern, F., Ning, G., Hong, Y., Khor, C.C., *et al.* (2015). Cloning and variation of ground state intestinal stem cells. *Nature* *522*, 173-178.
- Werth, M., Schmidt-Ott, K., Leete, T., Qiu, A., Hinze, C., Viltard, M., Paragas, N., Shawber, C., Yu, W., Lee, P., *et al.* (2017). Transcription factor TFCEP2L1 patterns cells in the mouse kidney collecting ducts. *Elife* *6*.

Reviewers' Comments:

Reviewer #2:

Remarks to the Author:

The authors have addressed all my previous comments, and more, in a satisfactory manner, and I'm happy to recommend the current version of the manuscript. I congratulate the authors on a beautiful study and manuscript.

Reviewer #3:

Remarks to the Author:

The authors have adequately addressed my questions and have significantly improved the quality of the manuscript with their revisions. I support publication.

REVIEWERS' COMMENTS

Reviewer #2 (Remarks to the Author):

The authors have addressed all my previous comments, and more, in a satisfactory manner, and I'm happy to recommend the current version of the manuscript. I congratulate the authors on a beautiful study and manuscript.

We thank the reviewer for supporting the publication of our study. We also thank the reviewer for the compliments of our manuscript.

Reviewer #3 (Remarks to the Author):

The authors have adequately addressed my questions and have significantly improved the quality of the manuscript with their revisions. I support publication.

We thank the reviewer for supporting the publication of our study.